# Track, Inpaint, Resplat: Subject-driven 3D and 4D Generation with Progressive Texture Infilling

**Shuhong Zheng**[1,2] **Ashkan Mirzaei**[3*] **Igor Gilitschenski**[1,2]

[1]University of Toronto  [2]Vector Institute  [3]Snap Inc.

{shuhong, ashkan, gilitschenski}@cs.toronto.edu

## Abstract

Current 3D/4D generation methods are usually optimized for photorealism, efficiency, and aesthetics. However, they often fail to preserve the semantic identity of the subject across different viewpoints. Adapting generation methods with one or few images of a specific subject (also known as *Personalization* or *Subject-driven* generation) allows generating visual content that aligns with the identity of the subject. However, personalized 3D/4D generation is still largely underexplored. In this work, we introduce TIRE *(Track, Inpaint, REsplat)*, a novel method for subject-driven 3D/4D generation. It takes an initial 3D asset produced by an existing 3D generative model as input and uses video tracking to identify the regions that need to be modified. Then, we adopt a subject-driven 2D inpainting model for progressively infilling the identified regions. Finally, we resplat the modified 2D multi-view observations back to 3D while still maintaining consistency. Extensive experiments demonstrate that our approach significantly improves identity preservation in 3D/4D generation compared to state-of-the-art methods. Our project website is available at https://zsh2000.github.io/track-inpaint-resplat.github.io/.

## 1 Introduction

Improving on the personalization quality of 3D/4D generation is a core challenge to enable impact and improve user experience. Current generation methods, however, are mostly guided by text prompts [11, 46, 68, 79, 96, 107, 119] or single images/videos [51, 88, 123, 129] to determine the front-facing appearance of the generated assets. Although these methods provide users with a certain amount of control on the content of the generated scene, they fall short in delivering *identity-preserving* outputs that are desired for personalized generation. As illustrated in Fig. 1(a), the state-of-the-art 4D generation model L4GM [73] fails to preserve the identity for the side and back views in the generated 4D asset. In the given example, it results in a blueish tone on the originally occluded regions of the cat. These limitations highlight the need for methods to enable subject-driven, identity-preserving 3D/4D generation for personalized applications.

Identity preservation in 3D/4D generation, though tempting, is challenging to accomplish. In single image- or video-guided 3D/4D generation, the model has few cues to infer the appearance of unobserved viewpoints and is forced to hallucinate. One line of research adopts score distillation sampling [1, 5, 12, 26, 29, 59, 68, 91] to optimize the appearance of the novel viewpoints. However, the time-consuming optimization process prevents the paradigm from being widely used in real-world applications. Moreover, during optimization, the appearance and motions of the 3D/4D assets often become averaged out [2, 132], which can reduce the quality of the generated content. More recently,

---

[*]Work done while at University of Toronto and Vector Institute.

39th Conference on Neural Information Processing Systems (NeurIPS 2025).

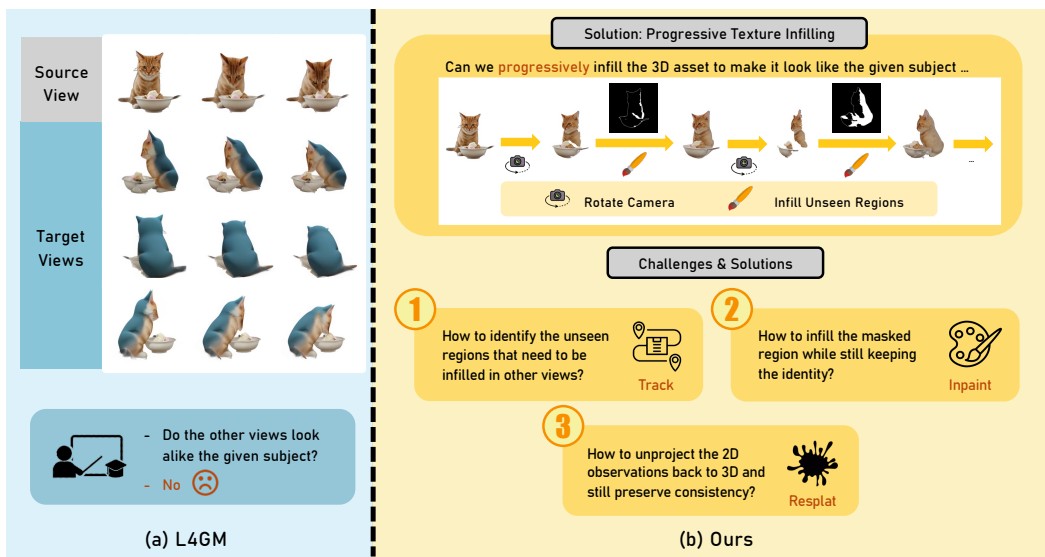

Figure 1: State-of-the-art 4D generation model L4GM [73] and our solution. **(a)** L4GM performs video-to-4D generation. The side and back views of the generated 4D asset does not look alike the subject in the given source view. **(b)** Our proposed solution TIRE *(**T**rack, **I**npaint, **RE**splat)* adopts the progressive texture infilling paradigm to inpaint the 3D asset to achieve subject-driven 3D/4D generation, which preserves the identity of the generated assets when observing from the novel views.

multi-view diffusion models are deployed in 3D/4D generation pipelines [54, 60, 78, 84, 92, 95, 106, 115] to hallucinate the appearance of a certain number of selected novel views. However, these models suffer from systematic errors in color and appearance for novel views. This is likely due to bias in the training data, which is difficult to address without careful dataset filtering. The most recent advancements [27, 42, 50, 102, 104, 110, 114, 124, 130] achieve superior efficiency with native 3D generation, *i.e.*, directly generating the 3D representation without per-scene optimization or multi-view observations as intermediate results. However, they still fail to produce results that can satisfactorily preserve the identity of the given reference as further demonstrated through the experimental analysis in Sec. 4.2 and App. F.

To effectively handle the challenges in 3D/4D generation for better personalization, we propose to perform subject-driven 3D/4D generation with progressive texture infilling, as shown in Fig. 1(b). Our proposed method, TIRE, is named after its three key components of the pipeline: ***T**rack, **I**npaint, and **RE**splat*. These three key components handle specific subtasks to achieve progressive texture infilling in a coordinated way: (1) *Track* identifies regions in other views that need infilling using long-video tracking. (2) *Inpaint* uses a customized 2D inpainting model to progressively infill unseen regions identified by *Track*, and ensures that the infilled content matches the subject's identity from the given source view. (3) *Resplat* unprojects the multi-view 2D infilled observations from *Inpaint* back to 3D while still maintaining consistency across multiple views. With these three components collaborating together in a cascaded manner, we achieve subject-driven 3D/4D generation while preserving the subject's identity.

We conduct extensive qualitative and quantitative evaluations, and find that our method serves as an effective general solution for enhancing the identity preservation in 3D/4D generation results upon the baseline methods. Moreover, our solution takes the exploration step in an *orthogonal direction* to current feed-forward approaches that utilize multi-view foundation models or native 3D/4D generation pipelines, which can be *complementary* to other advancements made in 3D/4D generation to collaboratively push forward the research field.

To summarize, our key contributions are threefold: First, we propose to solve subject-driven 3D/4D generation to enhance identity preservation of the generated 3D/4D assets for more personalized experience, which still remains a challenge for the most recent advancements on 3D/4D generation. It is an orthogonal while complementary effort towards high-quality generation from the foundation 3D/4D generation models. Second, to achieve the goal, we propose an innovative three-stage method,

TIRE: *Track, Inpaint, Resplat*. We first adopt video tracking for identifying regions that need infilling. Then, a customized 2D inpainting model is applied to infill the unseen regions. Afterwards, the 2D infilled observations are reprojected back to 3D to create the identity-preserving 3D/4D assets. Finally, comprehensive experimental results on our constructed DreamBooth-Dynamic benchmark and in-the-wild data showcase the superior performance of TIRE on subject-driven 3D/4D generation.

## 2    Related Works

**3D Generation.** Diffusion models [18, 80] have recently advanced and greatly improved 2D image/video generation [74]. To apply 2D diffusion model knowledge to 3D generation, Dream-Fusion [68] introduced score distillation sampling (SDS) to transfer knowledge to 3D. While some works [19, 25, 35, 39, 45, 69, 96, 99, 121] improved the quality and efficiency of optimization-based SDS, 3DiM [98], Zero-1-to-3 [51], and their successors [48, 49, 77] offered a different approach. They synthesized novel view images using diffusion models and then reconstructed 3D models from these synthetic views. Later works [7, 9, 10, 23, 28, 41, 52–55, 92, 97, 101] further enhanced the correctness and consistency of multi-view diffusion models. To accelerate generation, LRM [21] and concurrent works [40, 112] directly generate 3D representations using feed-forward networks. Subsequent works [86, 111, 126] applied this feed-forward approach to 3D Gaussians [36], a common 3D representation. Recent advancements [27, 100, 102, 110, 127] including MeshFormer [50], TRELLIS [104], and the Hunyuan3D series [114, 130] achieve fast native 3D generation that directly produce 3D representations after training on massive data. However, the goal of personalized customization is largely neglected as the field evolves, and even the most recent state-of-the-arts struggle to generate 3D content that satisfactorily preserve the identity.

**4D Generation.** Similar to 3D generation, early 4D generative methods [1, 2, 32, 72, 132] used a video version of SDS to transfer knowledge from video generation models to 3D space. Subsequently, multi-view video generation models [37, 108, 123, 125] were introduced. These models enabled 4D optimization using synthetic multi-view videos. Later research [31, 43, 44, 73, 82, 83, 94, 103, 109, 116, 118, 120, 122, 128] further improved the geometry, consistency, and efficiency of generated 4D assets. In contrast to efficiency-focused 3D/4D generation, we focus on subject-driven generation to preserve identity in generated assets, offering a more personalized option for 3D/4D content creation.

**Subject-driven Generation.** To personalize generated content for users, subject-driven generation became a prominent topic. Textual inversion [14] optimized a special text prompt token to represent a specific subject. DreamBooth [75] enabled text-to-image diffusion models to generate customized content for subjects by finetuning pre-trained models with a small number of example images. RealFill [89] extended this idea to subject-driven inpainting and adopted LoRA [22] for parameter-efficient finetuning. Subject-driven model personalization was also studied in multi-subject compositional generation [105] and videos [17, 30]. Beyond prior work in subject-driven 2D generation, DreamBooth3D [70] used image-to-image translation with personalized 2D models on multi-view observations to enhance identity in generated 3D objects. Customize-It-3D [24] proposed using a subject-specified prior to guide SDS optimization of generated assets. Make-Your-3D [47] finetuned a multi-view generation model with identity-aware optimization. Unlike previous works, we proposed leveraging the powerful 2D video tracking and inpainting tools to progressively infill the occluded regions of 3D/4D assets, preserving identity in the generated assets with the knowledge in 2D models.

## 3    Method

Given a single image or video of a certain subject, our goal is to generate a 3D (for images) or 4D (for videos) asset that faithfully represents the identity of this specific subject. Our proposed method, TIRE, consists of three stages: Track, Inpaint, Resplat. *Track* aims at providing the masks indicating the infilling regions from other viewpoints beyond the given source view (Sec. 3.2). *Inpaint* targets at progressively infilling the unobserved regions in other viewpoints with the infilled contents preserving the identity, while the regions are identified by the previous *Track* step (Sec. 3.3). *Resplat* is responsible for unprojecting the 2D infilled observations back to 3D (Sec. 3.4).

Our algorithm starts from a rough 3D/4D representation generated by existing models, as shown in the leftmost part in Fig. 2 about the setup stage. We render multi-view observations from the viewpoints within azimuth angle $\pm 180°$ and elevation angle $0°$, following the practice in [73, 86].

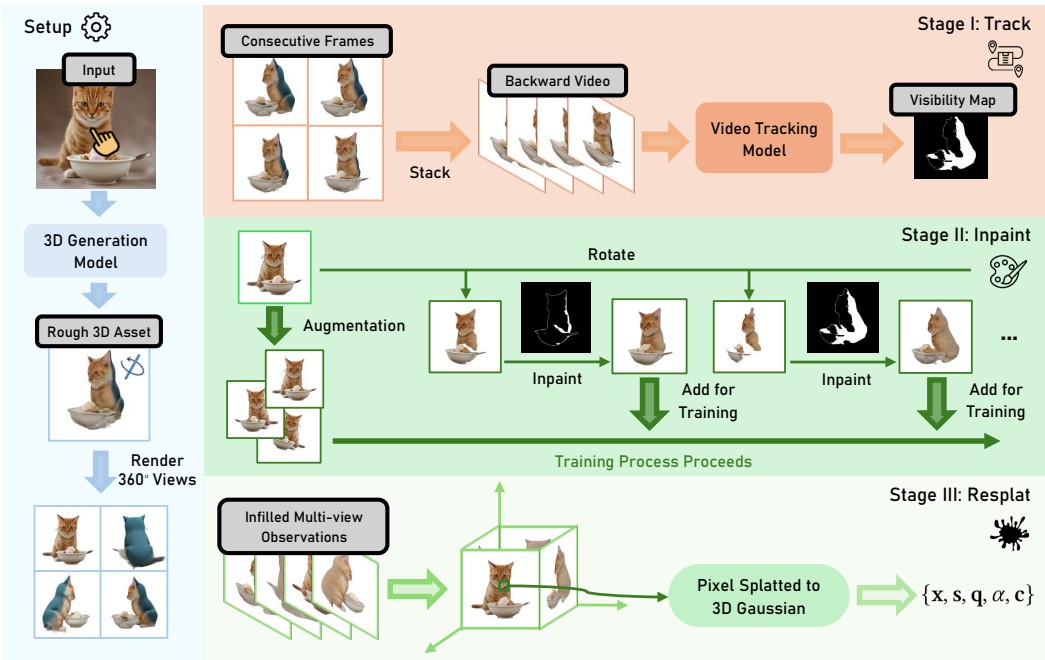

Figure 2: Pipeline of TIRE. TIRE starts from a rough 3D asset created by existing models and its rendered multi-view observations. Afterwards, the three stages *Track, Inpaint, Resplat* target at identifying the inpainting masks, infilling the occluded regions, and unprojecting back to 3D, respectively.

These initial rendered results will be used in our three-stage pipeline for infilling mask calculation, inpainting, and unprojection as discussed in the following subsections.

## 3.1 Preliminary

**Diffusion Models** [18] are generative models that learn a given data distribution by learning a denoising process that starts with random (typically Gaussian) noise and transforms this noise over multiple steps to the data distribution. In order to reduce the computational cost of learning distributions of image data, latent diffusion models [74] perform the diffusion and denoising processes in latent space. Pretrained autoencoders are used for the conversion between latent and image spaces.

In the diffusion process, we convert a clean latent $z_0$ to a noisy latent $z_T$ of arbitrary timestep $T$ as

$$z_T \sim q(z_T|z_0) = \mathcal{N}(z_T; \sqrt{\bar{\alpha}_T}z_0, (1-\bar{\alpha}_T)\mathbf{I}), \tag{1}$$

where the notation $\alpha_T = 1 - \beta_T$ and $\bar{\alpha}_T = \prod_{s=1}^{T}\alpha_s$ simplify the formulation with $\beta_T$ representing the schedule of the strength of the noise added in timestep $T$. When $T \to \infty$, $z_T$ is close to being equivalent to sampling from an isotropic Gaussian distribution.

The denoising process is the inversed operation to the diffusion process. The denoised latent at timestep $t-1$ can be estimated with the latent at timestep $t$ by

$$p_\theta(z_{t-1}|z_t) = \mathcal{N}(z_{t-1}; \mu_\theta(z_t, t), \boldsymbol{\Sigma}_\theta(z_t, t)), \tag{2}$$

where the parameters $\mu_\theta(z_t, t)$, $\boldsymbol{\Sigma}_\theta(z_t, t)$ of the Gaussian distribution are estimated from the diffusion model. As revealed in [18], $\boldsymbol{\Sigma}_\theta(z_t, t)$ only has negligible contribution on the results from the experiments. Therefore, the main objective of the denoising framework is to estimate $\mu_\theta(z_t, t)$, which is reparameterized with

$$\mu_\theta(z_t, t) = \frac{1}{\sqrt{\alpha_t}}\left(z_t - \frac{\beta_t}{\sqrt{1-\bar{\alpha}_t}}\epsilon_\theta(z_t, t)\right), \tag{3}$$

where $\epsilon_\theta(z_t, t)$ is the denoising network to predict the added noise $\epsilon$ for $z_t$ at timestep $t$.

The training objective for the denoising network is

$$\mathcal{L} = \mathbb{E}_{\epsilon \sim \mathcal{N}(0,1),t} \left[ \|\epsilon - \epsilon_\theta(z_t, t)\|_2^2 \right].$$ (4)

With the well-trained denoising network $\epsilon_\theta(z_t, t)$ and the deterministic sampling schedule in DDIM [81], the denoising process can be represented as

$$z_{t-1} = \sqrt{\alpha_{t-1}} \left( \frac{z_t - \sqrt{1 - \alpha_t}\epsilon_\theta(z_t, t)}{\sqrt{\alpha_t}} \right) + \sqrt{1 - \alpha_{t-1}}\epsilon_\theta(z_t, t).$$ (5)

**Large Reconstruction Models (LRM)** [21] are foundation models for 3D reconstruction that predicts triplane representation in a feed-forward manner. Later, large Gaussian model (LGM) [86] achieves feed-forward prediction from multi-view observations to 3D Gaussians, inspired by the previous works [6, 85] that utilize a U-Net to splat each pixel into a 3D Gaussian in space. To be more concrete, taking 4 multi-view images as input, LGM functions as

$$f : \mathbb{R}^{4 \times H \times W \times 3} \to \mathbb{R}^{\left(4 \times \frac{H}{2} \times \frac{W}{2}\right) \times 14},$$ (6)

where the output is a total of $\left(4 \times \frac{H}{2} \times \frac{W}{2}\right)$ 3D Gaussians, each has 14 parameters representing the center position $\mathbf{x} \in \mathbb{R}^3$, scaling $\mathbf{s} \in \mathbb{R}^3$, rotation quaternion $\mathbf{q} \in \mathbb{R}^4$, opacity $\alpha \in \mathbb{R}$, and color $\mathbf{c} \in \mathbb{R}^3$ of the 3D Gaussians. Follow-up work L4GM [73] extends the feed-forward 3D Gaussian generation to 4D with the additional time dimension, generating *a sequence of 3D Gaussians* that are consistent across both the spatial and time dimension.

## 3.2 First Stage: Track

**Goal.** As shown in Fig. 1(b), our proposed method adopts progressive texture infilling to achieve identity-preserving 3D/4D generation results. The first subtask we need to solve is to identify the regions that need to be infilled in viewpoints not observed in the original data. Obtaining a decent mask for infilling, however, poses non-trivial challenge.

**Identifying Infilling Masks with Video Tracking.** With the multi-view observations rendered from the setup stage, we can stack the consecutive frames together in the order of the camera movement to form a video. Then, we leverage the video tracking model CoTracker [34] to find the correspondence between the source view and the target views. The underlying principle is that, if the tracking result of a point on the target view is still within the valid mask of the target image, and with its visibility flag still active, we consider the point on the source view to be visible on the target view. An intuitive implementation is to start tracking from the given view, and see how the valid 2D pixels get propagated to the target views. However, as shown in the first row in Fig. 3, many small inpainting regions appear in the infilling mask, resulting in suboptimal grainy inpainting performance. Although this issue may be mitigated by performing other post-processing operations like dilations on the mask, it would require case-specific parameters

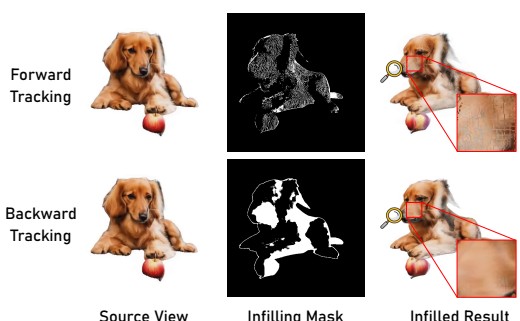

Figure 3: Comparison between forward tracking and backward tracking when identifying the inpainting mask. Forward tracking, which means that the tracking process starts from the given source view to the target views, though being more intuitive, leads to grainy inpainting results. In contrast, backward tracking produces more accurate masks in better shapes, which benefits the following inpainting process.

to control the post-processing operations. Instead, we design a wiser approach to obtain the mask with *backward tracking*, which performs video tracking from the target views to the given source view. As displayed in the second row in Fig. 3, the masks obtained with backward tracking is more accurate and also more suitable for the following inpainting process. The insight behind backward tracking is that the given source view contains the richest information about the subject's identity. Therefore, we start tracking from the target views to establish as many correspondences as possible with the given view, to effectively leverage the subject information present in the given source view. Our proposed solution is a model-agnostic approach that is generally applicable to any types of 3D representations, and also leverages the power of 2D models that are trained on massive video data.

## 3.3 Second Stage: Inpaint

**Challenges.** After obtaining the infilling masks for novel views in the *Track* stage, we need to wisely inpaint these regions to maintain the identity of the subject. We are facing two substantial challenges: **(1)** How to faithfully preserve the identity in the infilling regions; **(2)** How to inpaint the viewpoints that are far away from the given source view, as the reference appearance in the source view may be very different and is unable to provide direct guidance. To address challenge **(1)**, inspired by RealFill [89], we propose to personalize the pretrained stable diffusion [74] inpainting model to be subject-driven, aiming to preserve the identity of the inpainted region. To handle challenge **(2)**, we perform the inpainting process in a *progressive* manner, as we first start inpainting from the viewpoints that are close to the given source view. As the training proceeds, the model progressively learn to inpaint the viewpoints that are farther away from the original source view.

**Solution.** With the pretrained stable diffusion inpainting model in hand, we inject LoRA [22] weights in the pretrained model and finetune with randomly generated binary mask $m_i$ for inpainting. The loss calculation will only be conducted on the valid regions $m_v$ as

$$\mathcal{L} = m_v \odot [\epsilon_\theta(x_t, t, p, m_i, (1 - m_i) \odot x) - \epsilon], \tag{7}$$

where $m_v \in \{0, 1\}^{H \times W}$ is the valid mask in which 1 indicates the foreground of the image, while 0 indicates the background of the image. The valid mask is obtained by the same background removal tool used in recent 3D/4D generation works [54, 73, 86]. $p$ is a fixed language prompt *"A photo of sks"*. "$\odot$" denotes the element-wise multiplication and therefore $(1 - m_i) \odot x$ is the masked image. The other notations follow the same convention as Eq. 4.

At the beginning of the tuning process, since only a single image/video from the source view is available, we perform horizontal flipping and small-scale rotations within $15°$ on the original image for data augmentation. After training the inpainting task on the original image together with its augmented counterparts, we perform inpainting of azimuth angle $\theta = \pm 20°$ which we name it a *sweet spot*. This viewpoint serves as an *anchor viewpoint* for later processes when inpainting the viewpoints that are farther away from the source viewpoint. To be more concrete, when inpainting the farther viewpoints within $\pm 90°$, the similar operation of *backward tracking* in Sec. 3.2 will be applied to track from the queried viewpoints to this anchor viewpoint. It helps further reduce the area that needs to be infilled, therefore lowering the difficulty for the model to inpaint the far away viewpoints. The reasons for choosing $\pm 20°$ as the sweet spot for the anchor viewpoint in tracking are **(1)** compared to the source viewpoint at $0°$, it has certain exploration on unseen regions that can provide larger known regions for the farther viewpoints; **(2)** when inpainting the $\pm 20°$ viewpoint, since it does not have significant shift from the original training image, the inpainted results are relatively decent and reliable. Therefore, this *sweet spot* strikes a balance between *"exploration and exploitation"* of the given source view observation. After the $\pm 90°$ viewpoint is inpainted, it serves as the next anchor point for inpainting the rest of the viewpoints within $\pm 90° \sim \pm 180°$. As we do not want significant change on the original structure, we perform denoising with the first 30% of the denoising schedule following similar practice as [33].

## 3.4 Third Stage: Resplat

The goal of this stage is to unproject the inpainted 2D observations back to 3D. As the frames are infilled separately during the *inpaint* stage in Sec. 3.3, there may exist inconsistency across the inpainted frames. Thus, before lifting to 3D, we propose to use the multi-view diffusion model [92] to refine the consistency of the multi-view observations. More specifically, inspired by the previous work on mask-aware image editing [62], our multi-view denoising process only updates the latents on the unseen viewpoints as

$$z_{t-1} = \tilde{z}_{t-1} \odot M + \hat{z}_{t-1} \odot (1 - M), \tag{8}$$

where $z_{t-1}, \tilde{z}_{t-1}, \hat{z}_{t-1} \in \mathbb{R}^{(V+1) \times c \times h \times w}$. $\tilde{z}_{t-1}$ is the predicted latent at $T = t - 1$ during the denoising process obtained by Eq. 5. $\hat{z}_{t-1}$ is the noisy latent obtained by Eq. 1 with the forward diffusion process. $V = 4$ is the number of views, while $c, h, w$ are the channel number, height and width of the latents. As the multi-view diffusion model is image-conditioned, there are $(V + 1)$ entries on the first dimension of the latents, as the additional one being the latent of the conditional image. $M \in \{0, 1\}^{(V+1) \times c \times h \times w}$ is the mask for refinement with value 0 for the source view and 1 elsewhere. Similar to the practice of the *Inpaint* stage, only the first 30% of the denoising schedule

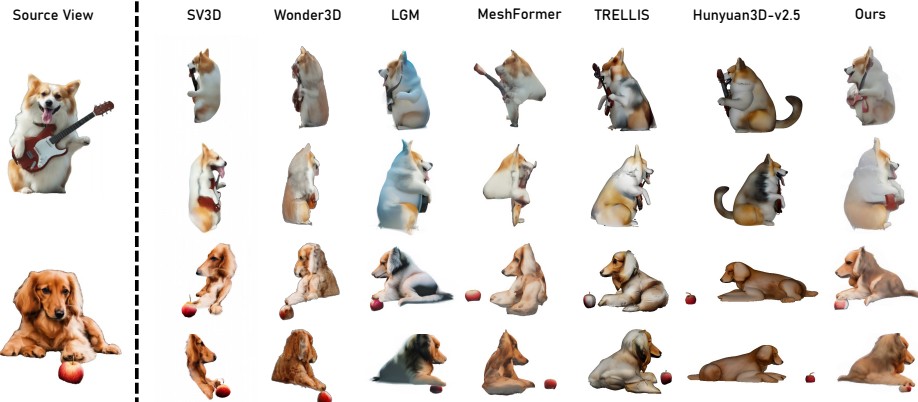

Figure 4: Qualitative comparison on image-to-3D generation with SV3D [90], Wonder3D [54], LGM [86], MeshFormer [50], TRELLIS [104], and Hunyuan3D-v2.5 [38]. Compared against other method in the image-to-3D setting, our method better preserves the identity of the reference image, and also reaches superior quality on geometry. It is noticeable that even for the most recent advancements in image-to-3D like TRELLIS and Hunyuan3D-v2.5, the challenge of producing identity-preserving 3D assets is still not well solved.

is applied. The mask-aware latent update strategy reinforces the identity preservation of the front view of the final 3D/4D assets. After refining the multi-view observations with Eq. 8, we *resplat* pixels in every viewpoint to Gaussians with [73, 86]. Note that this multi-view-to-Gaussian process is also adaptable to different large reconstruction models, and thus can be applied to different 3D representations beyond 3D Gaussians.

## 4 Experiments

### 4.1 Experimental Settings

**Datasets.** Starting from the original DreamBooth [75] dataset that focuses on subject-driven image generation, we construct a *DreamBooth-Dynamic* dataset. It is based on animatable subjects in the original DreamBooth dataset and will be used for subject-driven image-to-3D and video-to-4D generation. More details about the dataset are provided in App. B in the appendix. Besides the qualitative and quantitative evaluations on the constructed DreamBooth-Dynamic dataset, we also demonstrate that our method works for *in-the-wild data* that are displayed on the official project webpage of L4GM [73] in App. F in the appendix due to space issue.

**Baseline Methods.** We select Wonder3D [54], SV3D [90], LGM [86], MeshFormer [50], TREL-LIS [104], and Hunyuan3D-v2.5 [38] for the comparison on the image-to-3D task. For video-to-4D, we select recent 4D generation methods STAG4D [123] and SV4D [108][1], along with L4GM [73]. As few existing works focus on subject-driven 3D/4D generation [24, 47, 70], we choose Customize-It-3D [24] for comparison, as it is the most recent open-source option.

### 4.2 Qualitative Evaluation

We show visualizations of our results and the compared methods in Fig. 4 for image-to-3D generation and Fig. 5 and Fig. 6 for video-to-4D generation, rendered from different viewpoints and timesteps. From the qualitative comparisons, we observed the following:

**Identity-preserving Appearance.** Fig. 4 and Fig. 5 demonstrate that the 3D/4D assets generated by our method achieves significantly better identity preservation compared to the baselines. Notably, even the most recent advancements in image-to-3D generation, such as TRELLIS [104] and Hunyuan3D-v2.5 [38], still face substantial challenges in producing identity-preserving 3D assets, demonstrating that the task of subject-driven 3D/4D generation is a valid and important problem to investigate, highlighting the necessity of addressing it to achieve better identity preservation for 3D/4D generation.

---

[1]Comparison included in App. F in the supplementary material due to different pose settings.

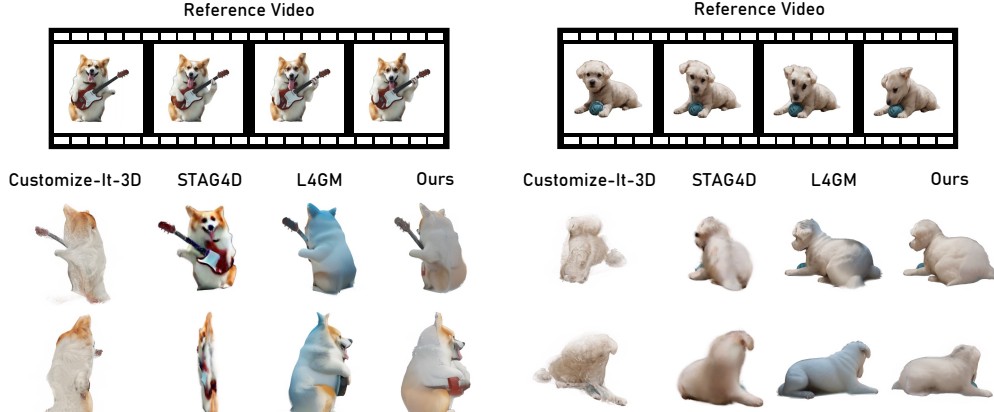

Figure 5: Comparison between our method and the baselines Customize-It-3D [24] (additional feed-forward operation from L4GM is applied after obtaining multi-view observations to allow it to generate dynamic 3D assets), STAG4D [123], and L4GM [73].

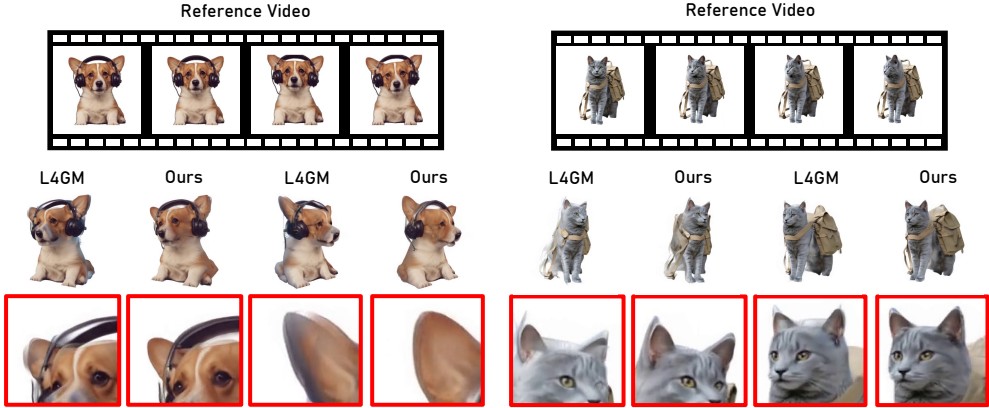

Figure 6: Comparison between our method and L4GM [73]. Although the main objective of our TIRE is to preserve the identity of the generated assets, the geometry of the generated assets also gets improved because the three stages in our pipeline collectively promote cross-view consistency.

**Enhanced Quality on Geometry.** Although TIRE mainly aims to improve subject-driven appearance modeling of the generated 3D/4D assets, it also demonstrates better geometry quality than the baseline methods. As shown in Fig. 6, our results outperform L4GM on the geometry of the generated assets. The rendered viewpoints from our method have fewer ghosting artifacts, which are caused by cross-view inconsistency. The reason that TIRE can fix the geometry is that the *Track* and *Inpaint* stages collaboratively propagate pixels seen in source views to the target views. Also, the *Resplat* stage includes a mask-aware halfway diffusion-denoising process in Eq. 8. These designs within TIRE implicitly refine the consistency of the multi-view observations and the generated 3D/4D assets.

**General Solution to 3D/4D Generation Methods.** As discussed in Sec. 3.4, the demonstrated results are generated with LGM [86] and L4GM [73] as the base model for our method. As our progressive texture infilling process only needs to perform on rendered 2D frames for tracking and inpainting, our method has the advantage of being generally applicable to all types of 3D/4D generation methods, no matter what representation the method adopts. To support this, we also include the results of how our TIRE can further enhance identity preservation for even the most advanced models like Hunyuan3D-v2.5 [38] in App. F.

### 4.3 Quantitative Evaluation

As there are no standard evaluation protocols for benchmarking the quality of subject-driven 3D/4D generation, we first follow [75] to adopt DINO [4] feature similarity to measure the subject fidelity of

the generated assets. More specifically, the source view is the reference image of the subject, and without loss of generality, 4D generation is selected as a case study.

For each generated asset, we render every 10 timesteps from 8 views around the assets, with a 45° interval in azimuth angle, to calculate the DINO similarity score with the reference image.

Table 1: Quantitative comparisons on the DINO feature similarity metrics from DreamBooth [75]. Best: **bold**, 2nd best: *italics*.

| Method | DINO (ViT-S/16) (↑) | DINO (ViT-B/16) (↑) |
|---|---|---|
| Customize-It-3D [24] | **0.5773** | **0.6087** |
| SV4D [108] | 0.5213 | 0.5426 |
| STAG4D [123] | 0.5287 | 0.5592 |
| L4GM [73] | 0.5506 | 0.5694 |
| TIRE (Ours) | *0.5665* | *0.5815* |

We report the results in Tab. 1, which is the average similarity score across all rendered images in all the scenes. Although TIRE outperforms L4GM and STAG4D in subject fidelity according to the DINO similarity metric, which is commonly used in the literature [13, 75], it is surprising that Customize-It-3D ranks highest. However, the qualitative comparisons in Sec. 4.2 clearly show that Customize-It-3D performs worse than other methods. This suggests that the original DINO similarity metric, while being a standard metric for subject fidelity, is not the most suitable quantitative measure for the nuances of the subject-driven 3D/4D generation task. We further discuss this limitation in App. C in the appendix.

Having recognized the challenges of conducting quantitative evaluations with purely vision-based models, we turn to vision-language models (VLMs) for more comprehensive assessment across multiple dimensions including identity preservation and visual quality. Specifically, we follow DreamBench++ [66] to adopt VLM-based evaluation on the identity preservation quality of the generated assets. Following the subject similarity evaluation option in DreamBench++, we ask the VLMs to give an overall score from 0-4 (0 is the worst, while 4 is the best) on subject consistency between the reference image and the generated images rendered from the assets, considering the following aspects: shape, color, texture, and facial features. Detailed implementations including the used prompts for VLMs are described in App. D. For the choice of VLMs, besides GPT-4o [64] which is adopted in DreamBench++, we also use five other VLMs (OpenAI o4-mini [65], Gemma 3 27B [15], Gemini 2.0 Flash [16], Qwen2.5-VL-7B [3], Mistral-Small-3.1-24B-Instruct [63]) with different architectures and sizes to foster the reliability of the evaluation. For each generated asset, we evenly render 8 views for each generated asset for evaluating the quality. The results in Tab. 2 show that our method achieves the best subject consistency when taking multiple aspects of consistency into account, demonstrating the effectiveness of our method for enhancing identity preservation for the generated 3D assets. Nevertheless, we can observe that the scores for all the methods still remain distant from the perfect score of 4, indicating that the task of subject-driven 3D/4D generation is still far from being solved.

Table 2: Quantitative comparisons on the VLM-based scores on the views rendered from the generated assets from different methods. Best is marked as **bold**.

| Methods | VLM-based scores (↑) | | | | | | |
|---|---|---|---|---|---|---|---|
| | GPT-4o | OpenAI o4-mini | Gemma 3 27B | Gemini 2.0 Flash | Qwen2.5-VL-7B | Mistral-Small-3.1-24B-Instruct | Average |
| TRELLIS | 1.332 | 1.426 | 1.870 | 1.402 | 1.596 | 1.228 | 1.476 |
| Hunyuan3D-v2.5 | 1.614 | 1.690 | 2.098 | 1.533 | 1.780 | 1.501 | 1.703 |
| TIRE (Ours) | **1.777** | **1.834** | **2.103** | **1.793** | **1.880** | **1.739** | **1.854** |

## 4.4 User Study

To further calibrate the identity preservation quality with human perception, we conduct user study on the 4D assets generated by Customize-It-3D [24], L4GM [73], and our method. We ask the volunteers to score the generated assets in a scale of 1-10 on the overall quality, where the participants are guided to focus on subject fidelity, cross-view consistency, *etc.* that are considered important factors for 3D/4D generation quality. We randomly select 10 samples from the DreamBooth-Dynamic dataset. Note that we *do not* explicitly tell the participants that we are

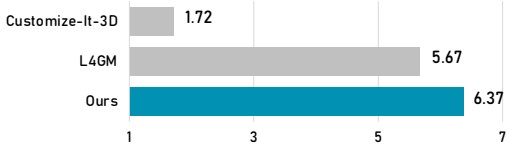

Figure 7: Results of our user study. Our method scores the highest in overall quality *without* explicitly informing the users that we are focusing on subject-driven generation.

working on improving subject-driven generation,

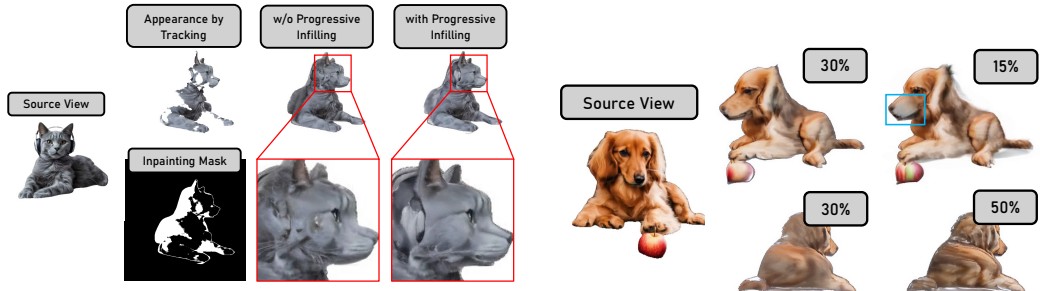

Figure 8: Ablation study on the progressive learning strategy in TIRE. Without adopting progressive learning, the model tends to consistently infill the appearance of the given source view *regardless of the current pose*, which results in wrongly infilling the textures on the side and back views.

Figure 9: Ablation study on different denoising schedules for inpainting. A smaller schedule may leave some regions unchanged, while a larger one may overly distort the textures.

which fosters the fairness in user study and reduces the underlying bias of the subjective preference towards our method. Details on the instructions and interface of our user study can be referred in App. A in the appendix. With 18 volunteers participating in the user study, we collect a total of 540 scores from participants. The results of the user study are reported in Fig. 7, which shows that our method is more subjectively preferred from the user experience.

### 4.5 Ablation Study

**Progressive Texture Infilling** is a core idea of our pipeline to infill the texture of unseen regions in the generated assets. Therefore, to prove the validity of our progressive infilling strategy, we demonstrate the ablation results illustrated in Fig. 8. More specifically, for the variant "w/o Progressive Inpainting", we use the inpainting model finetuned only on the original image of the source view and its augmented counterparts. When the progressive infilling strategy is not applied, the model tends to inpaint the appearance corresponding to the given source view of the object. We can observe that the model fills in textures of a cat's face with a pair of cat whiskers in the target view, even if the target view is $60°$ off the source view and actually displays the side view of the cat.

**Denoising Schedule for Inpainting.** As discussed in Sec. 3.3, we empirically adopt the first 30% of the denoising schedule in the *Inpaint* stage. To provide a more intuitive demonstration for guiding real practice, we examine the impact of a larger or smaller denoising schedule in Fig. 9. We can observe that when we adopt a smaller denoising schedule of 15%, some of the regions where the textures need to be refined remain unaltered, like the regions in the blue bounding box. On the other hand, when we choose a larger denoising schedule of 50%, we may have the risk of having an overly intense change on the textures, such that the appearance becomes less realistic.

Due to the limit of space, more ablation study can be found in App. E in the appendix.

## 5  Conclusions

We present TIRE, which aims at subject-driven 3D/4D generation that progressively infill the textures of the occluded regions. TIRE contains three important steps: *Track, Inpaint, Resplat. Track* helps identify the regions that need to be infilled via video tracking tools. *Inpaint* aims at infilling the occluded regions while preserving the identity of the subject. *Resplat* unprojects the multi-view infilled observations back to 3D with cross-view consistency. Extensive experimental results demonstrate that the 3D/4D assets obtained from our method achieve superior performance on both *more identity-preserving appearance* and *refined quality on geometry*. We believe that our solution is an important exploration on the direction complementary to the current research on efficient feed-forward 3D/4D generation pipelines, and can serve as a useful tool for expressing the creativity and enhancing the personalization on subject-driven 3D/4D generation. More discussions can be referred in App. C in the appendix.

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

# Track, Inpaint, Resplat: Subject-driven 3D and 4D Generation with Progressive Texture Infilling

## Technical Appendices and Supplementary Material

In the appendix, we first provide more details on the instructions and interface used in our user study in App. A to show how we foster the fairness and soundness of our user study. Next, in App. B, we illustrate the pipeline of how we construct our dataset for subject-driven 3D/4D generation. In App. C, we carry out more detailed discussions on how our proposed TIRE can *co-exist* and *co-develop* with other 3D/4D generation methods, the insights of our method design, together with the limitations of our approach. Then, more ablation study is presented in App. E. Afterwards, we provide additional qualitative results in App. F on in-the-wild data, more state-of-the-art advancements on 3D generation, *etc.* Moreover, in App. D, we elaborate on more implementation details to better support the reproducibility of the proposed method. Finally, the societal impact of our work is discussed in App. G.

## A   User Study

Below we present more details of our user study setup. We use the following instructions for participants at the start of the user study.

---

*For each scene, we use three different methods to generate 4D assets, which are rendered in the forms of videos and placed in random orders. We also provide a reference image to show the appearance of the subject. We would like to invite you to give an overall score from 1 to 10 to measure the quality of the generated results.*

*There are a few points to consider when providing the scores:*

- *Whether the videos are consistent across different views*

- *Whether different viewpoints of the generated results look alike the subject in the given reference image*

- *The visual quality of the videos (e.g., whether it has blurry artifacts, whether the rendered views look realistic)*

---

From the above instructions, note that we *do not* explicitly tell the participants that we are improving on subject-driven 4D generation. Instead, we ask the users to rate the generation results based on the overall quality. It fosters the fairness in user study and reduces the underlying bias of the subjective preference towards our method. We also include screenshots of our user study interface in Fig. A and Fig. B.

## B   More Details on Datasets

We show an overview of our dataset construction pipeline in Fig. C. More specifically, we train a customized text-to-image generation model on each subject following DreamBooth [75], with the backbone being an SDXL [67] model finetuned with LoRA [22]. The link of the dataset is https://github.com/google/dreambooth, and the license is Creative Commons Attribution 4.0 International (CC-BY-4.0). Then, we use manually created prompts to generate images with the subjects performing specific activities. Afterwards, we adopt the image-to-video model CogVideoX [20, 117] to create an animated video capturing the scene. Altogether, we curate a total of 23 animable subjects in the original DreamBooth dataset performing diverse activities.

## User Study on 4D Generation

Thanks for participating on the user study of 4D generation!

For each of the scene, we use three different methods to generate 4D assets, which are rendered in the forms of videos and placed in random orders. We also provide a reference image to show the appearance of the subject. We would like to invite you to give an overall score from 1 (worst quality) to 10 (best quality) to measure the quality of the generated results.

There are a few points to consider when providing the scores:

- Whether the videos are consistent across different views
- Whether different viewpoints of the generated results look alike the subject in the given reference image
- The visual quality of the videos (e.g., whether it has blurry artifacts, whether the rendered views look realistic)

There are 10 questions in total, and the estimated time for finishing this user study is 5-6 mins.

Figure A: Screenshot of the instructions of our user study. From the instructions, we *do not* explicitly tell the participants that our project is targeting at subject-driven 4D generation (*e.g.,* in the title we only mention "User Study on 4D Generation"). Instead, the users are asked to rate the generated results based on the overall quality, which enhances the fairness and reduces the underlying bias of the potential subjective preference towards our method.

## C   Discussions and Limitations

**Position of our proposed TIRE relative to existing models.** Existing state-of-the-art 3D/4D generation models [21, 23, 50, 58, 102, 104, 110, 114, 127, 130] are mostly feed-forward models with native 3D generation to achieve superior generation efficiency. Therefore, it is no denying that their feed-forward generation schema pioneers the "foundation models" for 3D/4D generation. Per-scene optimization-based methods [56, 93, 96, 113, 120, 131], do suffer from the limitation of efficiency as discussed in the training efficiency part in this section. However, we consider our method as an orthogonal effort towards subject-driven generation which is more customized. From the perspective of customers, it is valuable to have an option, supported by our method, to further preserve the identity and refine the generated results produced by the other feed-forward 3D/4D generation methods.

From a methodology perspective, some may argue that, to resolve the issues like the blueish shade shown in Fig. 1 in the main paper, we could heuristically design specific rules to filter out problematic data before training our model. However, it is infeasible to enumerate all the issues that may lead to data bias during the data filtering stage, as the issue shown in Fig. 1 is only one specific example of the many underlying issues that undermine the robustness of the feed-forward generation models. For example, if we place the rendered views of the generated assets from a specific model side by side, as shown in Fig. D, it becomes obvious to see that each model suffers from a specific bias pattern for side views and back views. For L4GM, the generated assets suffer from the unrealistic blueish or whitish color patterns similar to the case in Fig. 1. For SV3D, there are whitish tone and blurry texture at the generated viewpoints, which may originate from the imbalance object properties of the dataset used for training the model, or the bias from the network design. For Hunyuan3D-v2.5, although the shape and color tones looks generally reasonable, the appearance is over-smoothed and looks unrealistic. This could be still the consequence of using certain rules for data filtering and preprocessing, and the rest of the data gets biased towards the opposite direction. Therefore, it is extremely challenging and laborious to design data filtering principles to address the issue.

**Insights and innovations of the proposed TIRE pipeline.** Our solution is not a simple concatenation of existing methods, as seamlessly combining these works (some of them like video tracking methods are even seldom used in 3D generation) itself is already a non-trivial task. Our insight for devising the TIRE paradigm is that we can convert the problem of progressive inpainting for 3D assets to 2D problems, with *Track* and *Inpaint* identify and fill the occluded regions step by step with 2D

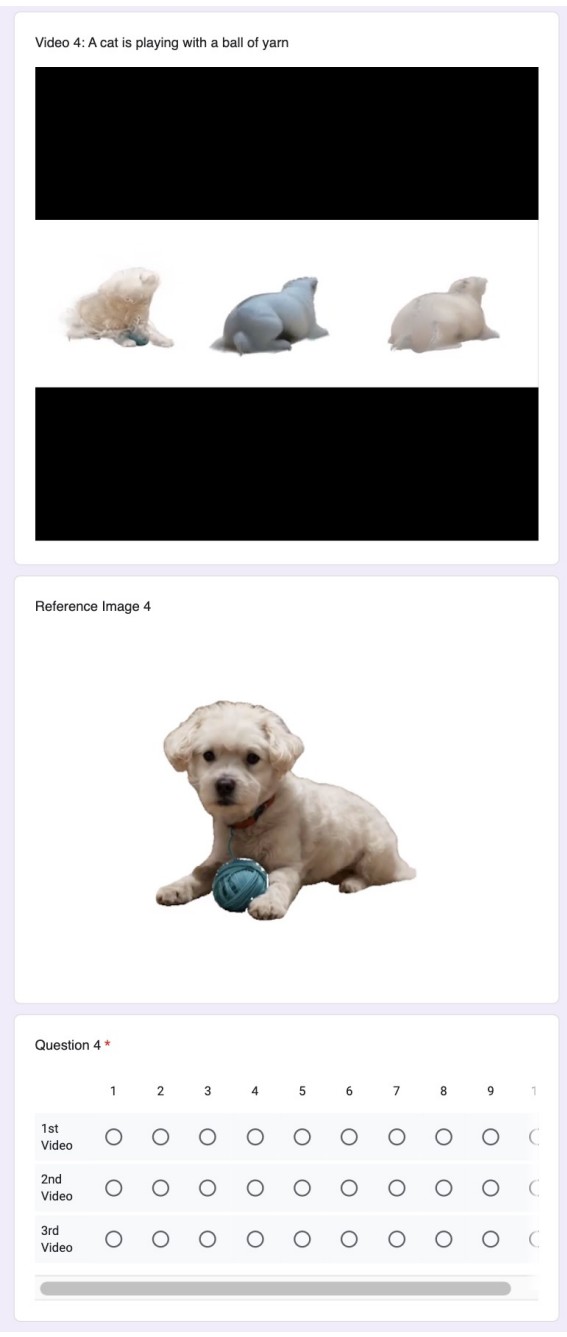

Figure B: Screenshot of an example question, containing videos generated by three methods placed side by side in random order, the reference front view, and the questions to score the three generated results.

models, followed by *Resplat* to fix the consistency in 3D. As a result, our pipeline is able to leverage the powerful 2D foundation models that are trained on substantially abundant and more diverse 2D data. Also, compared with other inpainting-based texture completion methods like Text2Tex [8] and InTeX [87], our pipeline can perform on any 3D representation including the implicit representations like neural radiance fields [61], while Text2Tex and InTeX require the adopted 3D representation to support surface modeling. Moreover, the three-stage pipeline is well orchestrated in a logical cascade. For example, the *Track* stage not only identifies the regions that need to be inpainted, but also

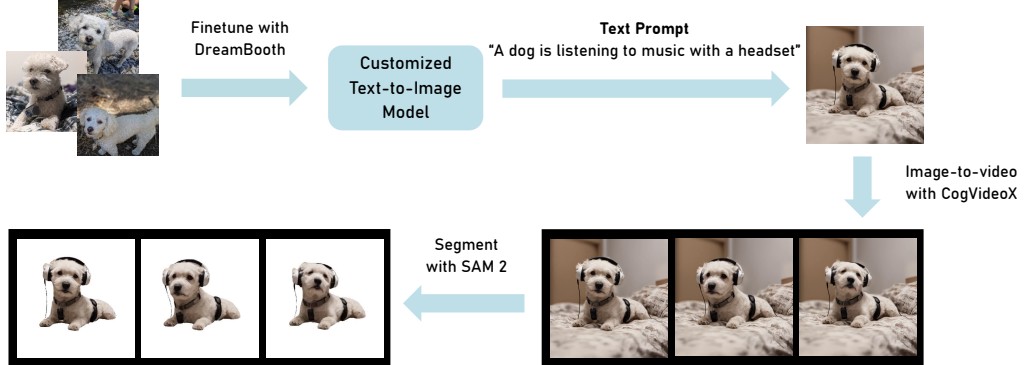

Figure C: Pipeline of constructing data for our *DreamBooth-Dynamic* dataset. First, a DreamBooth-finetuned [75] customized text-to-image model is trained on several casually collected images of a subject. Afterwards, we use manually created text prompts to generate images with the specific subjects. Afterwards, a powerful image-to-video model [117] is applied to animate the static images into videos. Finally, we use SAM 2 [71] to segment the foreground subjects to form the source view videos used in our evaluation process.

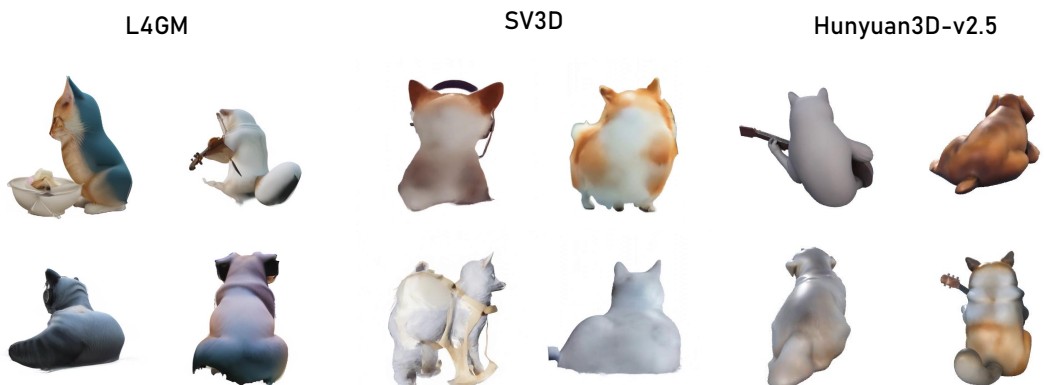

Figure D: Side and back viewpoints generated by specific models. It is obvious to see each model suffers from a bias pattern for generating the appearance of the occluded viewpoints.

propagate the corresponding pixels along the tracking path to novel views to mitigate the difficulty of the following *Inpaint* stage.

**Quantitative metrics for 3D/4D subject-driven generation.** As shown in Sec. 4.3 in the main paper, the quantitative metric adopted in DreamBooth leads to unreasonable results. We visualize an example in Fig. E to demonstrate the failure case of the DINO similarity metric. We can observe that the compared methods Customize-It-3D and STAG4D have obviously incorrect geometry for this viewpoint which is $135°$ away from the source view. However, as their shape or appearance looks similar to the reference which is the source view image, they score significantly higher than L4GM and our method, which at least correctly display the back view of the subject. Therefore, we believe that currently there are limitations in the quantitative evaluation on the subject fidelity of 3D/4D generation. A better evaluation metric for subject-driven 3D/4D generation needs to be proposed for properly benchmarking the performance of the methods. Potential solutions could be integrating the geometric evaluation with the appearance evaluation that is conducted by the current DINO feature similarity metric. The geometric assessment may pertain to the structural characteristics of the foreground of the generated images. We leave the design of proper evaluation metrics on subject-driven 3D/4D generation as future work.

**Training efficiency.** Our method takes around 100 mins on a single NVIDIA A100 GPU (when considering the memory consumption, an NVIDIA Quadro RTX 6000 or RTX 4090 with 24GB

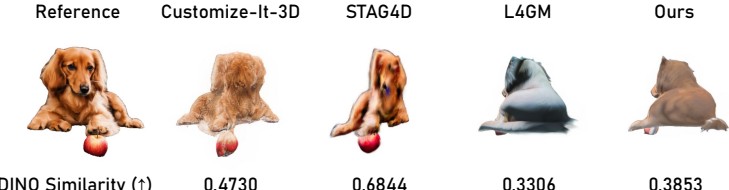

| | Reference | Customize-It-3D | STAG4D | L4GM | Ours |
|---|---|---|---|---|---|
| DINO Similarity (↑) | | 0.4730 | 0.6844 | 0.3306 | 0.3853 |

Figure E: An example of DINO (ViT-S/16) similarity scores of a rendered viewpoint which is $135°$ away from the source view across different methods. The baseline methods Customize-It-3D and STAG4D are obviously incorrect regarding geometry but get higher scores for DINO similarity.

will suffice), which is not very efficient. This issue is inherited from RealFill [89] which our method's pipeline is based on. However, since our solution focuses more on subject-driven 3D/4D generation, our work is an orthogonal effort to existing methods like L4GM [73], Turbo3D [23], *etc.* that work towards increasingly faster feed-forward 3D/4D generation pipelines. Moreover, personalization models are also in the trend of becoming feed-forward models [57, 76], without the need of finetuning on every subject like DreamBooth. Therefore, we believe that there will be more efficient subject-driven 2D inpainting models in the future that can greatly improve the efficiency of our approach.

**Why there are no specific operations designed for 4D generation in the temporal dimension?** Subject-driven generation, in the current context, mainly refers to the point that the appearance of the generated asset needs to conform to the reference image, which is often a static attribute. Therefore, temporal reasoning is not needed in most cases. Also, in our current implementation, the method L4GM [73] that we builds upon for 4D generation essentially generates a sequence of 3D Gaussians as mentioned in Sec. 3.1, which makes 4D generation, in the current situation, can be understood as the composition of a series of 3D generation processes. Nevertheless, we do acknowledge that there exist significant challenges specifically for 4D generation compared with 3D generation, and there are more complicated cases that customization in 4D generation also becomes important and especially challenging. For example, the reference objects shown in the given video may contain temporal patterns that change their appearance over time, and we want to generate 4D assets that faithfully preserve this property. We leave this as an interesting future direction to explore.

## D  Additional Implementation Details

### D.1  Training Details

As mentioned in the main paper, during the inpainting process, as we do not want to significant change the original structure of the observations from novel views, we perform denoising in the first 30% of the denoising schedule following similar practice as [33]. Therefore, the original structure of the images will not be destroyed after the inpainting process. Since we set the denoising schedule to only 30%, the original color of the images from the other views cannot be too deviant, which sometimes may not be satisfied with the original multi-view observations obtained by ImageDream. Therefore, we leverage another multi-view diffusion model Wonder3D [54], which empirically has inferior geometry but superior color than ImageDream, to provide guidance for color. Specifically, we splat multi-view observations from both ImageDream and Wonder3D to 3D Gaussians with LGM, and concatenate color features from Wonder3D and other features from ImageDream, followed by another halfway diffusion-denoising process with ImageDream. For progressive inpainting, except for the sweet spot $\pm20°$ and the secondary anchor viewpoint $\pm90°$, the other viewpoints are $\{\pm40°, \pm60°, \pm80°, \pm110°, \pm130°, \pm150°, \pm170°\}$. Each training stage consists of 300 training steps before advancing to the next stage. When adding the newly inpainted sample to the training data, we perform the same augmentation strategy when there is only one single source view image in the training set. We follow the other training parameters in RealFill [89] for finetuning the stable diffusion inpainting model (*e.g.,* batch size is 16, LoRA rank is 8, learning rate is 2e-4 for U-Net and 4e-5 for the text encoder). For the halfway diffusion-denoising process used after fixing the color of the rough assets with Wonder3D and before resplatting the pixels back to 3D, we find that the same denoising schedule of 0.3 as the inpainting stage generally works well in both cases.

## D.2 VLM-based Evaluation Details

We follow the concept preservation evaluation proposed in DreamBench++ [66] to prepare the prompts for VLMs to assign scores for the identity preservation quality between the reference image and the rendered view of the generated asset. The prompts are shown as follows.

---

### Task Definition

*You will be provided with an image generated based on reference image.*

*As an experienced evaluator, your task is to evaluate the semantic consistency between the subject of the generated image and the reference image, according to the scoring criteria.*

### Scoring Criteria

*It is often compared whether two subjects are consistent based on four basic visual features:*

1. *Shape: Evaluate whether the main body outline, structure, and proportions of the generated image match those of the reference image. This includes the geometric shape of the main body, clarity of edges, relative sizes, and spatial relationships between various parts composing the main body.*

2. *Color: Comparing the accuracy and consistency of the main colors generated in the image with those of the reference image. This includes saturation, hue, brightness, and whether the distribution of colors is similar to that of the subject in the reference image.*

3. *Texture: Focus on the local parts of the RGB image, whether the generated image effectively captures fine details without appearing blurry, and whether it possesses the required realism, clarity, and aesthetic appeal. Please note that unless specifically mentioned in the text prompt, excessive abstraction and formalization of texture are not necessary.*

4. *Facial Features: If the evaluation is of a person or animal, facial features will greatly affect the judgment of image consistency, and you also need to focus on judging whether the facial area looks very similar visually.*

### Scoring Range

*You need to give a specific integer score based on the comprehensive performance of the visual features above, ranging from 0 to 4:*

- *Very Poor (0): No resemblance. The generated image's subject has no relation to the reference.*

- *Poor (1): Minimal resemblance. The subject falls within the same broad category but differs significantly.*

- *Fair (2): Moderate resemblance. The subject shows likeness to the reference with notable variances.*

- *Good (3): Strong resemblance. The subject closely matches the reference with only minor discrepancies.*

- *Excellent (4): Near-identical. The subject of the generated image is virtually indistinguishable from the reference.*

### Input Format

*Every time you will receive two images, the first image is a reference image, and the second image is the generated image.*

*Please carefully review each image of the subject.*

### Output Format

*Score: [Your Score]*

*You must adhere to the specified output format, which means that only the scores need to be output, excluding your analysis process.*

---

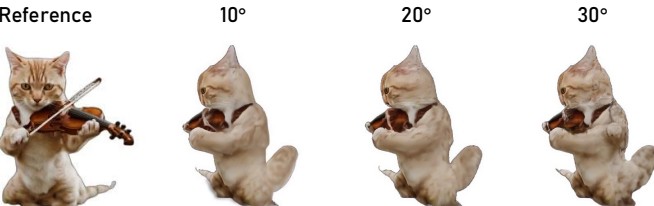

| Reference | 10° | 20° | 30° |
|-----------|-----|-----|-----|

Figure F: Ablation study on the degree of progressiveness during the *Inpaint* stage. Our current choice of degree of progressiveness (20°) strikes a balance between inpainting quality and efficiency. Smaller degree of progressiveness (10°) yields decent results but slows down the whole inpainting process, while larger degree of progressiveness (30°) brings noticeable degradation in the inpainting quality.

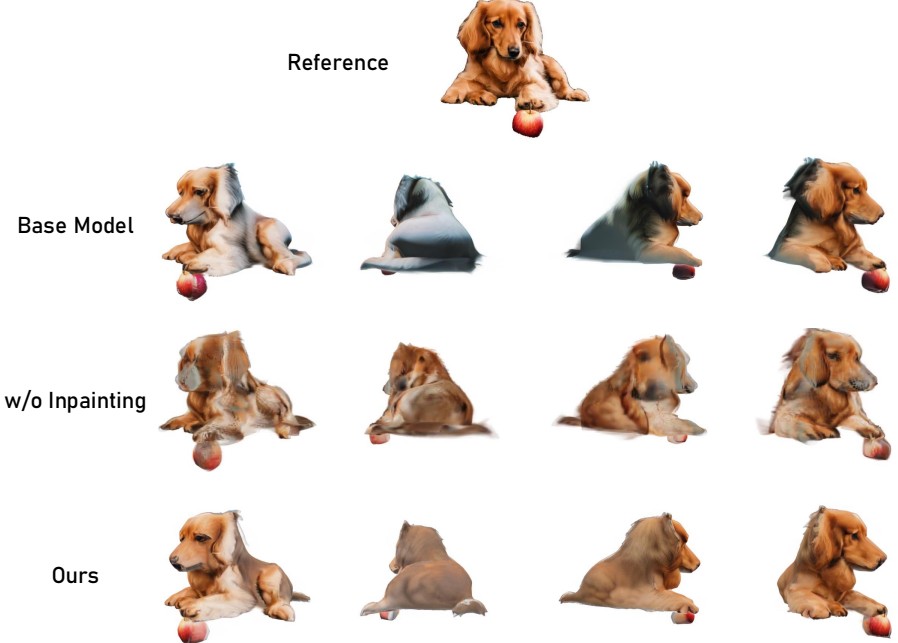

Figure G: Ablation study on the necessity of having the inpainting process. The color and texture of the assets without our inpainting process would be far from having satisfactory quality and decent identity preservation.

# E    More Ablation Study

## E.1    Degree of Progressiveness for Inpainting

We empirically choose 20° as the degree of progressiveness in the inpainting stage as described in Sec. 3.3 for striking a balance between the inpainting quality and the algorithm efficiency. We display the ablation on selecting a smaller or larger degree of progressiveness (10° and 30°) in Fig. F to show the effect of different choices on progressiveness. Basically, for the degrees of progressiveness smaller than the current choice, the visual results are very similar to the current results, but we need additional steps to calculate the inpainting masks for the increased number of training stages. For the degrees of progressiveness larger than the current choice, we could observe that the regions (especially for those that are far away from the reference view, e.g., the regions that are close to the back view for the target views of ±90°) are inpainted worse than the current implementation. Our current choice of 20° of progressiveness balances between the number of frames that need for tracking, and the difficulty of inpainting the unseen regions that are far away from the source view.

Table A: Quantitative comparisons the assets generated by Hunyuan3D-v2.5 [38] and our enhancement built upon it. Best is marked as **bold**.

| Methods | VLM-based scores (↑) | | | | | | |
|---|---|---|---|---|---|---|---|
| | GPT-4o | OpenAI o4-mini | Gemma 3 27B | Gemini 2.0 Flash | Qwen2.5-VL-7B | Mistral-Small-3.1-24B-Instruct | Average |
| Hunyuan3D-v2.5 | 1.614 | 1.690 | **2.098** | 1.533 | 1.780 | 1.501 | 1.703 |
| TIRE on Hunyuan3D-v2.5 (Ours) | **1.750** | **1.712** | 2.092 | **1.609** | **1.899** | **1.707** | **1.795** |

Table B: Quantitative comparisons the assets generated by L4GM [73] and our enhancement built upon it on the in-the-wild data displayed on the L4GM official website. Best is marked as **bold**.

| Methods | VLM-based scores (↑) | | | | | | |
|---|---|---|---|---|---|---|---|
| | GPT-4o | OpenAI o4-mini | Gemma 3 27B | Gemini 2.0 Flash | Qwen2.5-VL-7B | Mistral-Small-3.1-24B-Instruct | Average |
| L4GM | **2.743** | 2.441 | 2.640 | 2.022 | 2.104 | 2.125 | 2.346 |
| TIRE on L4GM (Ours) | 2.601 | **2.478** | **2.684** | **2.110** | **2.191** | **2.163** | **2.371** |

## E.2 Necessity of the Inpainting Process

As mentioned in App. D, we leverage another multi-view diffusion model Wonder3D [54] which empirically has superior color than the base 3D generation model LGM [86] that we mainly use in the paper, to roughly fix the color first before applying our method. We would like to show the necessity of our method by demonstrating how much identity is restored with our inpainting process. In Fig. G, we can see that without inpainting, there are many obvious flaws from the results. For example, on the right view of the scene, the dog's nose has a large grayish patch. The dog's body is also covered with messy dark brown patterns. Therefore, it is still far away from a good 3D asset for decent identity preservation. After the inpainting is done, the large grayish patch on the dog's nose gets infilled with reasonable color, and the textures on the dog's body gets more realistic and coherent with the given source view.

# F  More Experimental Results

For an overview of this section, first we show the comparison on in-the-wild data to show the robustness of our method in Sec. F.1. Then, we illustrate that our method is generally applicable to all types of 3D/4D generation methods by showing its application on one of the most recent image-to-3D methods. Next, comparisons with more state-of-the-art advancements are demonstrated in Sec. F.3 to show that recent advancements are generally not robust enough to produce personalized 3D assets that well preserve the identity of the subjects. Afterwards, more comparisons on image-to-3D and video-to-4D generation are presented in Sec. F.4. Finally, the comparisons with SV4D is displayed in Sec. F.5 due to different pose settings.

## F.1  Comparisons on In-the-wild Data

We additionally demonstrate the performance of our model on in-the-wild data, which are the examples displayed on the official webpage of L4GM [73], as their validation dataset has not been released. The comparison of our method against L4GM is shown in Fig. H and Tab. B. We can observe that our method more faithfully preserves the subject fidelity in the generated 4D assets and achieves superior overall quality compared with the baseline.

## F.2  General Solution to 3D/4D Generation Methods

Our solution is a *plug-in solution* that is generally applicable to all types of 3D/4D representations and methods. This benefit is a natural outcome from our method design of fully leveraging 2D tools to solve this 3D task. To apply on any 3D/4D methods, it is as simple as replacing the process of generating the initial assets in our pipeline with the 3D/4D generation that we want to improve upon. We show the improvement of our method upon the current state-of-the-art advancement in 3D generation Hunyuan3D-v2.5 [38] in Fig. I and Tab. A. We can observe that our method makes refinement to the texture of the generated 3D assets of Hunyuan3D-v2.5, yielding results that better match the identity of the reference images.

## F.3 Comparisons with More State-of-the-art Methods

We showcase the comparison with more state-of-the-art advancements in Fig. J, including Hunyuan3D-v1.0 [114], SPAR3D [27], and commercial models like Sudo AI[1] (built upon One-2-3-45++ [49]) and Neural4D[2] (built upon Direct3D [102]). Together with the results of MeshFormer [50], TREL-LIS [104], and Hunyuan3D-v2.5 [130] in Sec. 4.2, we can observe that even the most recent advancements in 3D generation still struggle with producing well-rounded personalized 3D/4D contents that well preserve the identity of the subject.

## F.4 More Qualitative Comparisons on 3D/4D Generation

We include more comparisons on image-to-3D generation with methods Wonder3D [54], SV3D [90], and LGM [86] in Fig. K. The comparisons demonstrate that our method outperforms existing image-to-3D approaches with better identity preservation and superior geometry accuracy. Additional visual comparisons on video-to-4D generation in Fig. L to demonstrate the *identity-preserving appearance* and the *enhanced quality on geometry* produced from our method, indicating the effectiveness of our approach.

## F.5 Comparisons with SV4D

We display the comparison with SV4D [108] in Fig. M. The reason that we show the SV4D results separately in the supplementary material is that the official code of SV4D has only released the multi-view video generation part, without the following 4D optimization step. Therefore, only a certain number of selected views can be rendered with its official code. For simplicity, the viewpoints selected for our method are close viewpoints but not the identical ones. However, we can still observe from the qualitative comparisons that SV4D fails to produce decent results regarding both texture and geometry.

# G   Societal Impact

We expect our work to have a meaningful and positive impact on the society. We sincerely wish that our method can ignite the creativity inside people to design personalized 3D/4D assets that serve versatile purposes. Also, we hope that our work highlights a critical and often overlooked challenge during the evolution of 3D/4D generation: the task of generating identity-preserved 3D/4D assets for personalized applications still remains unsolved. By bringing attention to this gap, we aim to raise awareness within the research community and encourage further exploration in this direction.

**Potential negative societal impact.** Our work is likely the same as other research on data generation regarding potential negative societal impact with the risk of digital forgery. Also, if the technique is mistakenly used, copyright and ethical issues may occur.

---

[1] https://www.sudo.ai/image-to-3d
[2] https://www.neural4d.com/studio/image-to-3d

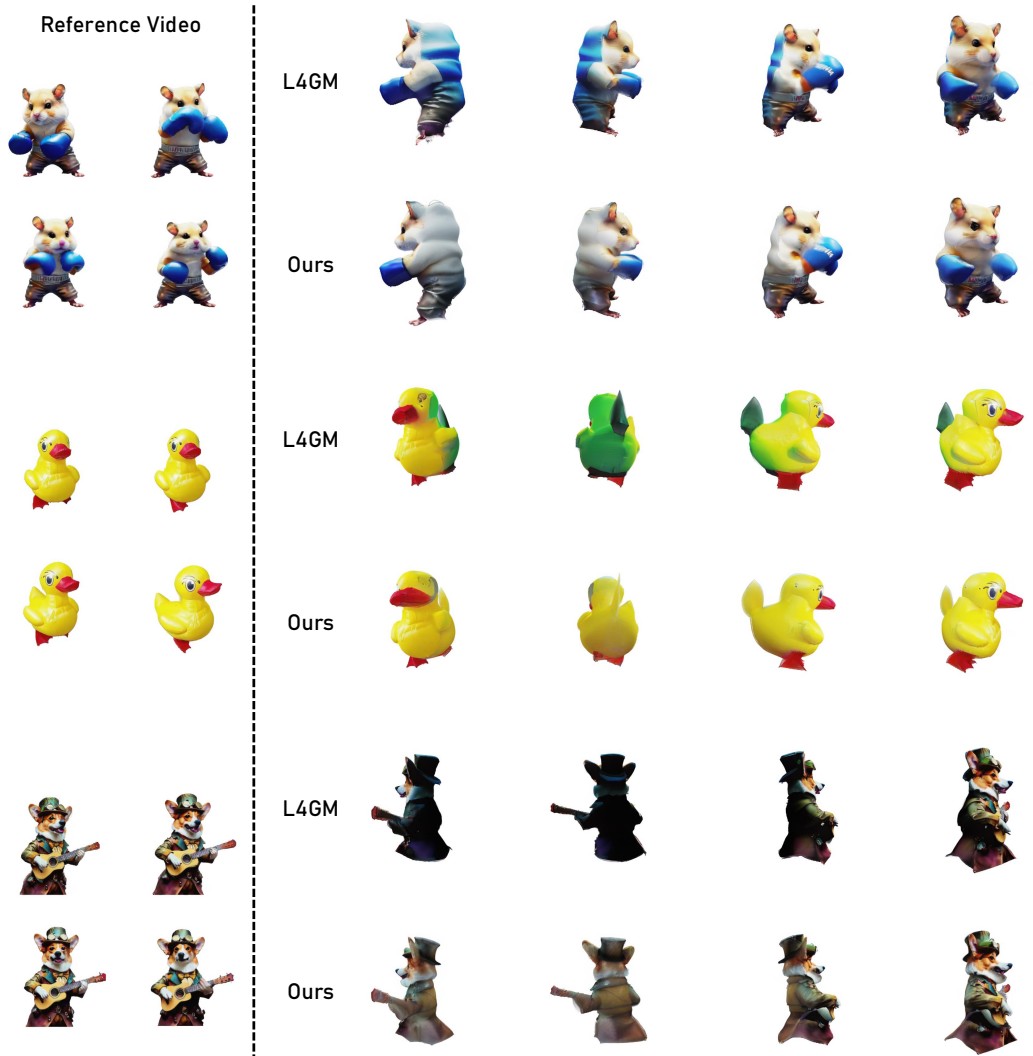

Figure H: Qualitative comparisons on the in-the-wild videos which are displayed on the official website of L4GM [73]. The visualizations demonstrate that our results more faithfully reflect the identity of the given subjects, yielding better overall quality.

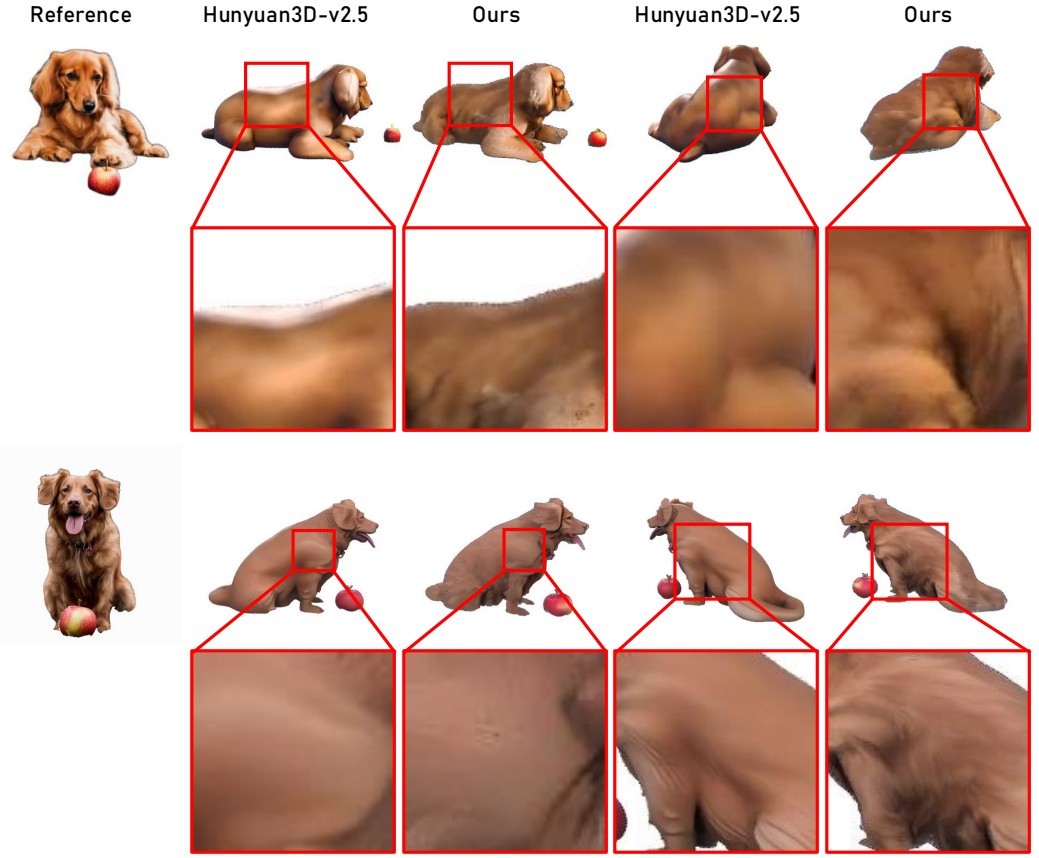

Figure I: Qualitative results showing the improvement of our method when applying on the most advanced image-to-3D method Hunyuan3D-v2.5 [38]. We can observe that our method produces superior texture of the generated assets which better match the identity of the reference image.

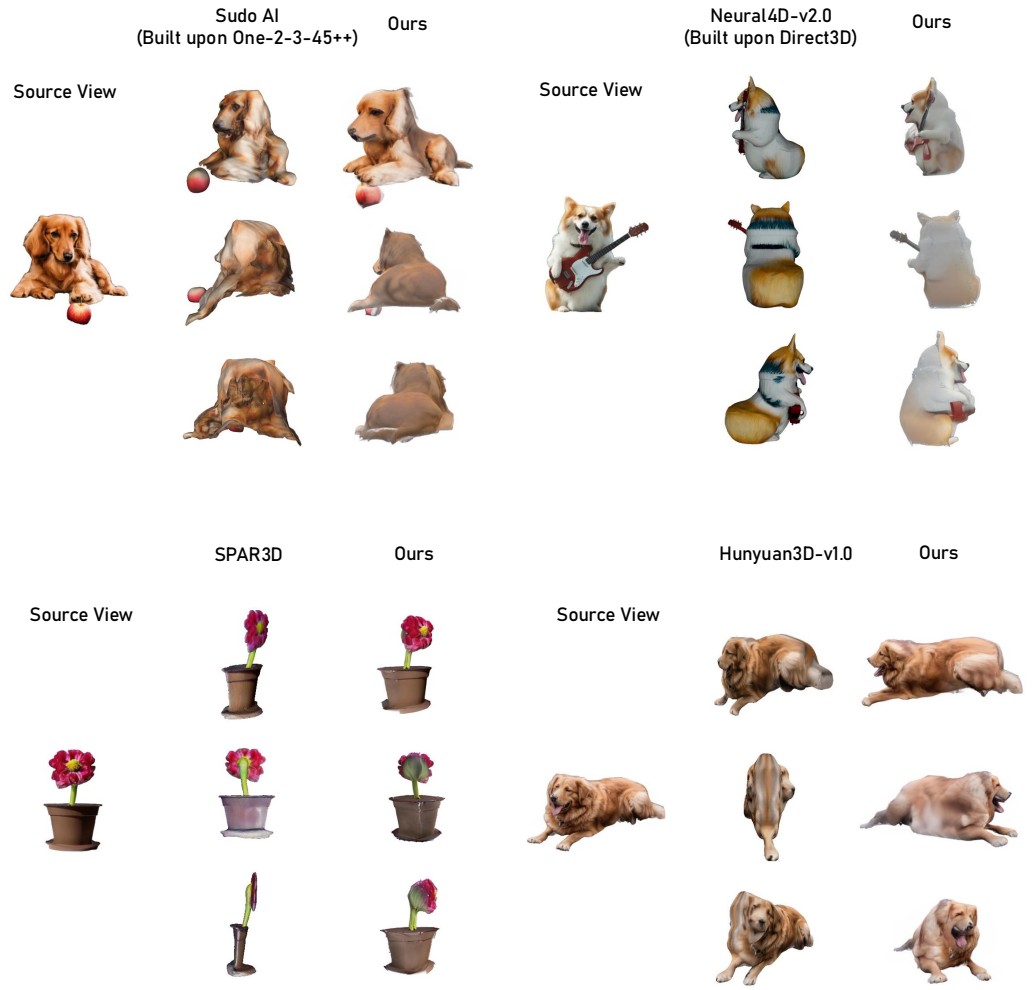

Figure J: Qualitative comparison with more advanced methods including Hunyuan3D-v1.0 [114], SPAR3D [27], Sudo AI (built upon One-2-3-45++ [49]), and Neural4D (built upon Direct3D [102]). Even the recent advancements in 3D generation cannot satisfactorily handle the personalized 3D generation challenge.

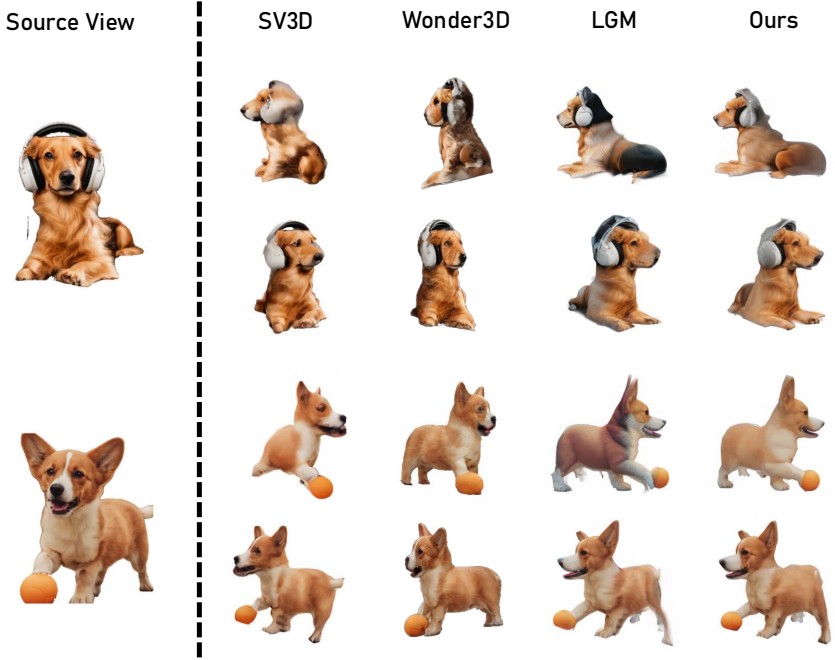

Figure K: Qualitative comparison with SV3D [90], Wonder3D [54], and LGM [86]. Compared against other method in the image-to-3D setting, our method achieves better preserves the identity of the reference image, and also reaches superior quality on geometry.

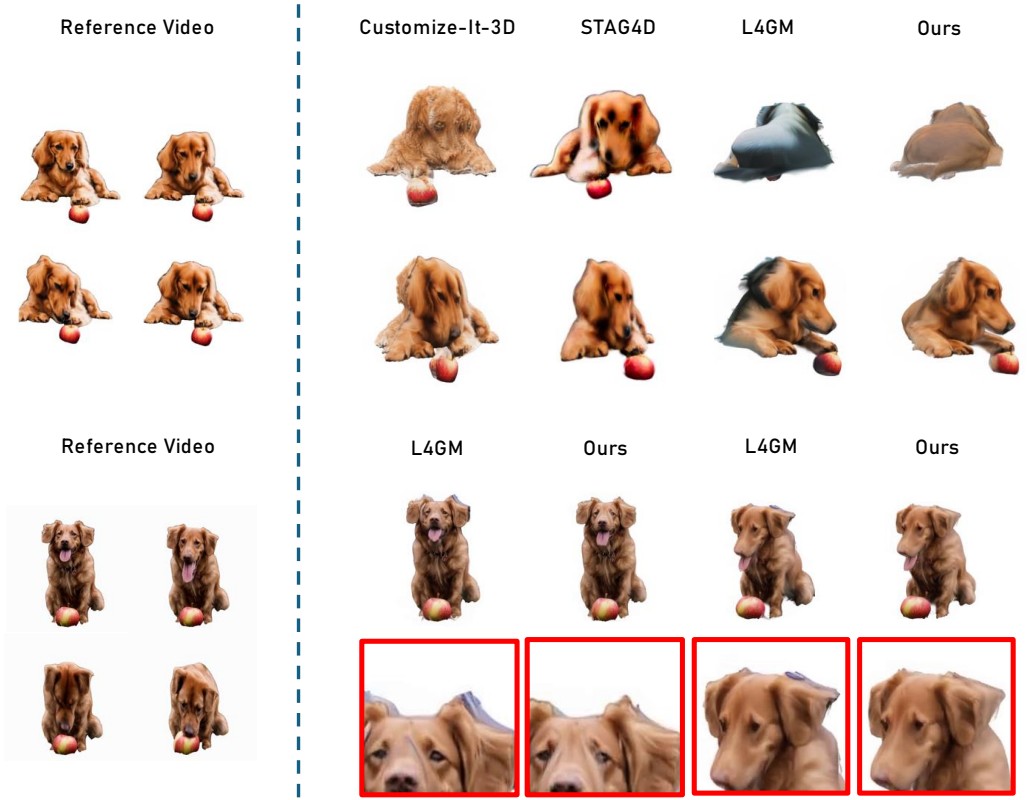

Figure L: Additional qualitative results of the comparison between our method and the baselines Customize-It-3D [24], STAG4D [123], and L4GM [73]. Compared with other methods, our solution achieves superior subject fidelity along with improved geometry on the generated assets, due to our design in our three-stage framework that fosters cross-view consistency.

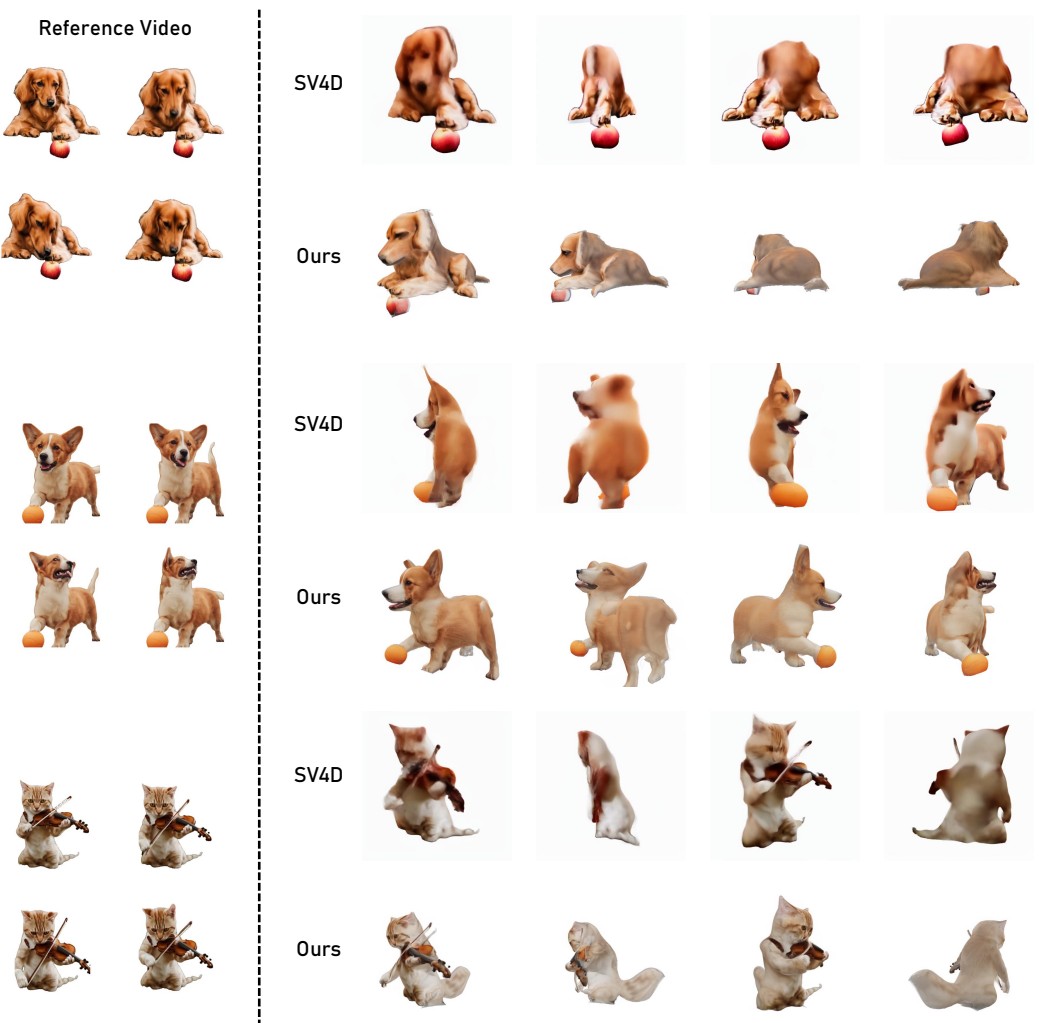

Figure M: Qualitative comparison with SV4D [108]. Since only a certain number of selected views can be rendered with the official code of SV4D, we choose the close viewpoints but not the identical ones for comparison for the sake of simplicity. Nevertheless, it is still obvious that SV4D has inferior performance compared with our method in the perspectives of both appearance and geometry.

