# OpenReview forum: "Track, Inpaint, Resplat: Subject-driven 3D and 4D Generation with Progressive Texture Infilling"
_NeurIPS.cc/2025/Conference — NeurIPS 2025 poster_

### Official Review · Reviewer_AB6n · 2025-07-01

**Clarity:** 3
**Significance:** 2
**Originality:** 2
**Rating:** 4
**Confidence:** 4

**Summary:**

TIRE (Track, Inpaint, REsplat)  method for 3D/4D generation with semantic consistency across views.
This method maintains the consistency of the target through three stages of processing. The tracking strategy adopts a backward tracking optimization scheme that supports any 3D representation form to improve tracking accuracy. The repair strategy uses a progressive repair strategy to solve the consistency problem when the target perspective and the source perspective differ too much. The final reconstruction stage uses multi-perspective consistency optimization to improve the visual consistency of the final result.

**Questions:**

- For tracking, which 3D expression works best and which works worst? Why? Please add some ablation experiments
- For tracking, backward tracking. What are the disadvantages of backward tracking, and what are the common situations where backward tracking fails and normal tracking succeeds?
- For the inpaint phase, it is necessary to explain the correlation between the current progressive design and loss design, and add experiments with different degrees of progressiveness.
- Please optimize the description of Figure 1 so that it does not appear to be an automatic annotation or determination method based on LLM.
-

**Ethical Concerns:**

["NO or VERY MINOR ethics concerns only"]

**Final Justification:**

The authors have provided a strong rebuttal along with additional experiments that will be included, so I am upgrading my rating to Borderline Accept.

**Limitations:**

- The multi-stage solution will accumulate errors. At the same time, there is no overall optimization loss or optimization goal between the stages, or in the end, resulting in the three stages being equivalent to three independent algorithms.
- The algorithm based on progressive rotation restoration is prone to misalignment in details and will have obvious segmentation lines. You can try to give some experimental results of leopards with spots or giraffes with more patterns.

**Paper Formatting Concerns:**

No Paper Formatting Concerns

**Quality:**

3

**Strengths And Weaknesses:**

Strengths:
- The three-stage approach solves several core contradictions in visual consistency, and creatively uses backward tracking to improve the effect.
- From the experimental results, compared with the existing SOTA 3D solutions (including closed-source solutions), it is indeed better in terms of visual consistency.
- In theory, this method can handle a variety of different 3D representations well.

Weaknesses:
- The experimental phase did not experiment with different 3D representations, so it is not possible to judge which representation the method is most suitable for and which representation it will have problems with.
- The method part does not explain much about backward tracking, and does not give the specific implementation of backward tracking and the problems that may arise (why a solution like cotracker can use backward tracking, and why backward tracking is used in the field of 3D generation)
- Figure 1 is somewhat misleading. In theory, the whole process can be automatically run in stages, but Figure 1 looks like it requires more human thinking and intervention.

---

> ### Author Rebuttal · Authors · 2025-07-31
>
> We sincerely appreciate the detailed feedback you provide for our work! Below, we provide the following clarifications to address your concerns:
>
> ---
>
> **W1 & Q1: Which 3D representations are more suitable?**
>
> Thanks for the valuable comment! Since our tracking is performed on rendered 2D images from the 3D assets, the performance relies on the quality of the rendered images, rather than solely depending on the type of 3D representations (e.g., NeRFs, 3D Gaussians, 3D mesh). Basically, better cross-view consistency of the generated 3D assets would lead to better performance in tracking, because they have less floating artifacts that can distract the tracking model.
>
> For the performance of the whole pipeline, it is better to adopt the 3D generation methods that can generate decent geometric shapes of the subject. The reason is that our pipeline is designed mainly to fix the unnatural appearance issue to make the generated assets match the identity of the given source view, with little modification on the geometry and shape of the subject (there are geometric fixes from our pipeline, as demonstrated in Fig. 6 in the main paper and Fig. I in the appendix, but they are not major changes to the geometry). Therefore, starting from a 3D method that can produce decent geometry and shape is more favorable for our method to achieve optimal performance.
>
> We have prepared interesting visual comparisons for ablations on different methods with different 3D representations. However, due to the policy of the rebuttal not allowing for PDF or other types of media, we choose to describe the qualitative comparisons in detail to the reviewer. *The qualitative comparisons requested for the following questions are addressed in the same way.*
>
> > ***Description of Visual Comparisons***
>
> > If solely focusing on the tracking performance, methods like Wonder3D with 3D mesh as representation produces slightly better tracking results than methods like LGM with 3D Gaussians and SV3D with NeRFs, because 3D mesh generally has fewer floating artifacts than NeRFs and 3D Gaussians, which makes it easier for tracking.
>
> > However, for producing a better final generated 3D asset, we empirically find that LGM has an overall better capability of producing realistic shapes and geometry of the subjects, better handling the depth of the subject along with the spatial relationship between the objects. Although the tracking performance on LGM may not be as accurate as Wonder3D (but not far away from Wonder3D), we observe that our method can offset the effect in the following *Resplat* stage when rectifying the multi-view geometry, generally being robust towards the minor inaccuracies from the tracking stage.
>
> Therefore, using LGM helps us yield the most pleasant results for the final generated 3D assets, which is the reason for our current implementation to use LGM as the initial 3D generation method in the pipeline.
>
>
> ---
>
> **W2 & Q2: Normal tracking v.s. backward tracking**
>
> Thanks for the question! In our method, backward tracking is to stack the frames from the target view to the source view to form a video, and then use the video tracking tool CoTracker to track the pixels in the target view. If the tracking result of the pixel gets lost or falls outside the valid regions of the source view, it means the pixel needs to be inpainted in the following *Inpaint* stage. The only difference between the videos for forward tracking and backward tracking is the camera movement direction. Therefore, the video tracking tool CoTracker can well handle the videos with reversed-order frames as normal.
>
> For comparing normal tracking (forward tracking) and backward tracking, backward tracking is almost always a better choice than normal tracking, both from the design principle and the experimental experience. For the underlying design principle of backward tracking, we can make an analogy to *image warping* to better understand this. There are two warping solutions: **(1)** Forward mapping, and **(2)** backward mapping (reverse mapping). In forward mapping, the algorithm iterates all the pixels in the source image $I_1$ to map their values to the target image $I_2$. However, this can lead to "holes" in the target image because many pixels in $I_2$ may not get mapped from a certain pixel in $I_1$. In comparison, when performing backward mapping, any pixel in $I_2$ that should be mapped to a certain pixel in $I_1$ can successfully find its corresponding pixel in $I_1$. The problem of backward mapping, in the image warping context, is that pixels in $I_2$ which are occluded or do not exist in $I_1$, cannot get its value during the mapping process. However, in our context of backward tracking, ***it is exactly what we are searching for!*** The goal of the *Track* stage is exactly to identify the pixels in the target view that are occluded or missing in the source view. Therefore, our backward tracking design perfectly matches our need for figuring out which parts in the target views need to be inpainted. Additionally, from our experimental experience (Fig. 3 as an example), backward tracking is indeed a more suitable solution for our pipeline, which matches the design principle of our backward tracking algorithm.
>
> ---
>
> **W3 & Q4: Figure 1 design**
>
> Thanks for the suggestion! We will revise the figure to remove the icon of the thinking person, and reorganize the components of the figure to better convey the broad picture of our method.
>
> ---
>
> **Q3: The correlation between the progressive design and loss design, and ablations on degrees of progressiveness**
>
> Thanks for the comment! If we understand correctly, the reviewer would like to know about the correlation between our adopted inpainting loss (Eq. 7) and the general diffusion-based inpainting loss in works like RealFill [a]. The loss calculation steps between these two are the same, and the difference is that we are performing an innovative ***progressive*** inpainting strategy. This is because our task poses additional challenges compared to the original RealFill setting, which requires the model to infill the viewpoints that have significant variance from the source viewpoint. If we only train the inpainting model with the source view, the model will not be competent enough to fill in the textures for the side and back views that have large variations.
>
> For the ablations on the degrees of progressiveness, we have also prepared informative visual comparisons for the reviewer, and the description of the qualitative comparisons is shown below:
>
> > ***Description of Visual Comparisons***
>
> > Basically, for the degrees of progressiveness smaller than the current choice (current choice is $20^\circ$, and we choose $10^\circ$ for ablation), the visual results are very similar to the current results, but we need additional steps to calculate the inpainting masks for the increased number of training stages. The total training time can roughly stay the same, because we can half the iterations needed for each stage and still get comparable results. For the degrees of progressiveness larger than the current choice (we choose $30^\circ$), we could observe that the regions (especially for those that are far away from the reference view, e.g., the regions that are close to the back view for the target views of $\pm 90^\circ$) are inpainted worse than the current implementation. Our current choice of $20^\circ$ of progressiveness balances between the number of frames that need for tracking, and the difficulty of inpainting the unseen regions that are far away from the source view.
>
> If the reviewer still has questions regarding the correlation between the current progressive design and loss design, please feel free to further clarify this question and we are happy to answer!
>
> [a] Tang et al. RealFill: Reference-Driven Generation for Authentic Image Completion. ACM TOG 2024.
>
> ---
>
> **L1: Error accumulation with the multi-stage solution**
>
> Thanks for the comment! It is true that multi-stage solutions have the disadvantage of error accumulation. However, our three stages are well orchestrated, with the earlier stage striving to make the task of the later stage easier. At the same time, the later stage is designed to make up for the errors which are caused by the inaccuracies from the earlier stages. For example, one of the purposes to use backward tracking is to produce inpainting masks that are more suitable for the following *Inpaint* stage. The *Resplat* stage refines the multi-view consistency with mask-aware latent updates to mitigate the error coming from the prior *Track* and *Inpaint* stages. Therefore, the three-stage solution is cohesively integrated with each other, instead of just functioning as three independent algorithms.
>
> ---
>
> **L2: Limitation on progressive rotation restoration**
>
> Thanks for the insightful comment! Subjects with complex patterns are indeed challenging for our current model, which are also the hard cases for other 3D generation models. However, we believe that our work can provide inspiration on how to utilize the powerful 2D tools like tracking and inpainting for solving this challenge. For the segmentation lines, it can be largely suppressed with using soft masks during inpainting. Basically, at the mask boundaries we apply feathering to blur the mask, then use the blurred soft mask for integrating the original image and the inpainted image. With this simple operation, the segmentation line issue can be significantly addressed.
>
>
> ---
>
>
> We sincerely thank you once again for your valuable comments, which have greatly helped improve our work. We hope that the explanation above can well address your concerns.

---

> > ### Author Response · Authors · 2025-08-05
> >
> > Dear Reviewer AB6n,
> >
> > We would like to once again thank you for your valuable review and thoughtful questions!
> >
> > As the discussion period is coming close to an end, we would like to send you a reminder about our rebuttal as above to solve your concerns. Please check whether your concerns have been addressed. We are sincerely looking forward to hearing from you, and are always happy to have more discussions with you!
> >
> > Thank you once again for your insightful reviews!
> >
> > Best,
> >
> > Authors

---

> > ### Comment · Reviewer_AB6n · 2025-08-05
> > **thanks for the response**
> >
> > Regarding W2, 3, and 4, I think the authors' responses are reasonable and convincing.
> > However, regarding W1, I agree that the authors' statement that LGM has the most suitable performance is the best. And, since the article states that this method is applicable to all 3D expressions, I still think that additional experiments should be added to the appendix to explore the advantages and disadvantages of this method under different expressions.

---

> > > ### Author Response · Authors · 2025-08-05
> > >
> > > Dear Reviewer AB6n,
> > >
> > > We are sincerely glad to see that you find our responses reasonable and convincing! For W1, we appreciate your agreement on our statement of which representations are more suitable. We will definitely add the informative visual comparisons *(mesh-based representation used in methods like  Wonder3D produces the best result in tracking, while methods like LGM with more decent shape and geometry are more suitable for producing final results)* in the appendix as suggested by you to show the advantages and disadvantages of different representations and methods, and also showcase that our method is generally capable of handling any 3D representation.
> > >
> > > We would like to thank you once again for your valuable suggestions, which greatly helps us to improve our work! If you find that all your concerns have been addressed, we would truly appreciate it if you could kindly consider raising the score!
> > >
> > > Best,
> > >
> > > Authors

---

### Official Review · Reviewer_HPe1 · 2025-07-03

**Clarity:** 2
**Significance:** 2
**Originality:** 2
**Rating:** 4
**Confidence:** 3

**Summary:**

The paper proposes a method for generating a 3D model from a single RGB image such that the identity of the subject is preserved. This is done in a progressive manner by first creating a coarse 3D representation using a pre-trained 2D-to-3D model; rendering this representation from novel viewpoints; tracking regions of the rendered views that are not visible in the source image; inpainting these regions using an identity-aware diffusion-based inpainter; and finally, combining all rendered+inpainted views into a single Gaussian representation after enforcing consistency among views.

**Questions:**

1. The paper does not provide any details about how the iniital 3D model is generated. In particular, what method is used to lift the image from 2D to 3D?
2. It is not clear to me how the inpainting model can be successfully finetuned on an identity with a prompt such as "a photo of sks" which does not include a class ("a photo of an sks dog/cat/vase") of the subject. Some more details and justification for the choices made here would be very helpful.
3. Why don't the authors compare all baselines in the quantitative evaluation stage? Specifically, the SOTA methods Trellis and Hunyuan3D are not included in Table 1.

**Ethical Concerns:**

["NO or VERY MINOR ethics concerns only"]

**Final Justification:**

The paper answered most of my concerns and have shown a clear path towards addressing the remaining questions I had.

**Limitations:**

Yes.

**Quality:**

2

**Strengths And Weaknesses:**

Strengths:
1. The method shows fairly good results using pre-trained and fine-tuned components.
2. The idea of fine-tuning an identity-preserving inpainting model for view synthesis is novel and interesting.

Weaknesses:
1. The motivation for the task the paper is solving, and what exactly "identity preservation" means is not clearly defined in the introduction. This is especially important since "personalization" and "identity preservation" are terms more commonly used by generative models which use text prompts to generate subjects in novel settings ("an sks dog in an astronaut suit"). But such text-based control doesn't seem to be present in the current model.
2. The qualitative evaluation is limited solely to pictures of cats and dogs. In these cases it is difficult to judge how effectively identity is preserved. Furthermore, the title of the paper implies applicability to general objects, not just animals. Therefore, this should be supported with additional qualitative results on diverse objects.
3. The paper claims to work for 4D reconstruction based solely on the assumption that all its steps translate to video. But no results on 4D reconstruction are shown. This claim should be toned down.
4. The evaluation section does not support the claims of the paper. Specifically, the qualitative results of the method, while good, are not clearly better than past work. In fact, they may arguably be worse than Trellis and Hunyuan3D. The quantitative results are also not better. If, as the authors claim, the quantitative metrics are incapable of capturing the improvement of the method, then they should explore or devise metrics that better capture the contribution of identity preservation. Perhaps the authors can refer to the "VBench" paper to get a comprehensive list of metrics for generative tasks. Furthremore, the user study does not ask participants to judge identity preservation, which is the main contribution of the paper.
5. Section 3.1 provides far too much detail about diffusion models that is not directly related to the task at hand. Diffusion models are common enough now for this information to be known beforehand by readers. If absolutely necessary, it should be moved to a supplemental. The additional space can be used for more qualitative results as suggested above.

---

> ### Author Rebuttal · Authors · 2025-07-31
>
> We sincerely appreciate the constructive feedback you have given for our work! Below, we provide the following clarifications to address your concerns:
>
> ---
>
> **W1: More clarification on the task setting**
>
> Thanks for the comments! It is true that our setting is a bit different from the traditional personalization setting of using generative models to generate subjects with novel text prompts. In comparison, our setting aims at generating 3D/4D assets that correspond to the identity of the subjects when observing from different views. Both settings are generating something of the specific subject in certain conditions. The traditional personalization setting is "under certain text prompts", while our setting is "under certain viewing directions", which is the reason for us to name our task "subject-driven 3D/4D generation". We will make it more clear in the final version of our paper.
>
> ---
>
> **W2: More diverse samples**
>
> We indeed have evaluations on in-the-wild data as mentioned in Lines 247-249, and the results are shown in Fig. F and the vase example in Fig. G (though only the vase is a non-animal object). The results seem to be biased towards animals because animals are the most common objects that are used for dynamic scenes, as for the examples shown in the L4GM website, animals also take the majority of their chosen objects. However, our work can definitely work on other subjects beyond animals. First, many of our examples are scenes of animals interacting with certain objects like apples and violins, which demonstrates that our method is applicable to other objects to certain extent. Second, we have prepared interesting visual comparisons and want to show the reviewer of how our method is capable of enhancing identity preservation for more general objects. Due to the policy of the rebuttal not allowing for PDF or other types of media, we choose to describe the qualitative comparisons in detail to the reviewer.
>
> > ***Description of Visual Comparisons***
>
> > We would like to describe two cases for demonstrating the capability of our method. Both cases are objects from in-the-wild images. The first case is a house covered with snow on the roof. One obvious issue for other 3D generation methods like SV3D and LGM is that they fail to preserve the property of the covering snow on top of the house, while our method is capable of inpainting the snow pattern on the roof from the back view. The second case is a traffic cone with red and silver stripes. Other 3D generation methods like SV3D and LGM mess up the stripe patterns and produce unnatural colors. With the assistance of our method, the texture and appearance of the traffic cone get well preserved.
>
>
> ---
>
> **W3: 4D results**
>
> We do have 4D reconstruction results in the paper. We would like to kindly point out to the reviewer about our 4D results as follows. Figs. 5, 6 in the main paper and Figs. F, I, J in the appendix show the snapshots of different timesteps from different views for the 4D assets. Also, the user study is conducted in the 4D setting, as presented in Sec. A in the appendix. Participants are shown with a series of videos showing the comparisons of the 4D scenes for scoring. For animable demonstrations, there are also examples in the video in the supplementary material (03:15 - 04:12) for reference. Therefore, we do have abundant 4D results and demonstrations for the paper.
>
> ---
>
> **W4-1: Evaluation on identity preservation**
>
> Thanks for the suggestion from the reviewer! After checking the VBench paper, we get inspired to use vision-language models (VLMs) for evaluating the identity preservation capability of the models. We follow the setting of DreamBench++ [a] for evaluation, and use a variety of VLMs with different architectures and sizes to ensure the reliability and soundness of the evaluation. The results are shown as follow:
>
>
> |Methods|GPT-4o ($\uparrow$)|OpenAI o4-mini ($\uparrow$)|Gemma 3 27B ($\uparrow$)|Gemini 2.0 Flash ($\uparrow$)|Qwen2.5-VL-7B ($\uparrow$)|Mistral-Small-3.1-24B-Instruct ($\uparrow$)|Average ($\uparrow$)|
> |:---|:---:|:----:|:----:|:----:|:----:|:----:|:---:|
> |TRELLIS|1.332|1.426|1.870|1.402|1.596|1.228|1.476|
> |Hunyuan3D-2.5 | 1.614 | 1.690 | 2.098 | 1.533 | 1.780 | 1.501| 1.703  |
> |TIRE (Ours) | **1.777** | **1.834** | **2.103** | **1.793** | **1.880** | **1.739** |**1.854**  |
>
> We can observe that **(1)** our method consistently outperforms the state-of-the-arts TRELLIS and Hunyuan3D-2.5 in the quantitative evaluation regarding identity preservation and generation quality; and **(2)** different VLMs share consistent scoring rank for different models, demonstrating the robust generalization of the adopted evaluation protocol.
>
>
> [a] Peng et al. DreamBench++: A Human-Aligned Benchmark for Personalized Image Generation. ICLR 2025.
>
>
> ---
>
> **W4-2: User study on identity preservation**
>
> Identity preservation is actually already taken into account in the user study. In the instruction of the user study, as shown in Sec. A in the appendix, we can see that there is the instruction *"Whether different viewpoints of the generated results look alike the subject in the given reference image"* (Lines 623-624) that informs the participants that whether the identity of the generated novel views match the given source view is a crucial factor for the scores. Therefore, we do ask the participants to focus on identity preservation, it is just we ***avoid to explicitly*** ask them to specifically focus on this point.
>
> The reason for us to not explicitly emphasize this point is that we want to foster the fairness of the user study. If we explicitly mention that our project is about identity preservation, the outcome will be obvious because our method is designed to fix this issue. It will make participants consciously feel in preference of our method. Therefore, our current user study setting is more fair, and can reflect the overall quality of all the methods.
>
> ---
>
> **W5: Diffusion model details in Section 3.1**
>
> Thanks for the suggestion! The original reason for us to have a relatively long Sec. 3.1 is because we want to present the equations for diffusion models, especially Eqs. 4 and 5, that are needed for reference when introducing the inpainting loss (Eq. 7) and mask-aware multi-view refinement (Eq. 8). We will definitely shorten the preliminary for diffusion models and present more interesting qualitative results in the main paper!
>
>
> ---
>
> **Q1: Initial 3D model generation**
>
> Thanks for pointing out! In our current implementation, we use LGM for lifting the image from 2D to 3D. Actually, the method for lifting from 2D to 3D can be arbitrarily chosen, because our tracking and inpainting infill the unobserved regions with only 2D models. Therefore, for any chosen method to lift the image from 2D to 3D, we can always render multi-view observations from whatever 3D representation is used, and apply our algorithm afterwards.
>
> ---
>
> **Q2: Prompts used during progressive inpainting**
>
> We thank the reviewer for the valuable question! The inpainting process can actually work well without the class name of the subject! The justification for us to choose this implementation is to follow the solution in RealFill [b], which demonstrates that the inpainting process is still feasible without having the object name or the scene description in the text prompts. We keep this setting in our work, so it serves as a more general and simpler solution for inpainting the scene. The reason for the model being able to inpaint without the guidance of the text prompt is that the special token $\hat{V}$ (or *sks*) is supposed to be optimized for capturing the content of the scene, instead of only representing the identity of the object. This is because the only task the model needs to focus on is to inpaint this specific scene, instead of generating the object in novel text prompts like the traditional setting mentioned in your previous comment of ***W1: More clarification on the task setting***. Therefore, the model still works well without mentioning the class name or concrete contents of the scene.
>
> Nevertheless, your comment is indeed very inspiring, as including the scene content in the text prompt would definitely have the potential to further improve the inpainting performance!
>
> [b] Tang et al. RealFill: Reference-Driven Generation for Authentic Image Completion. ACM TOG 2024.
>
> ---
>
> **Q3: Quantitative evaluation**
>
> The original quantitative evaluation in the paper is to showcase the limitation of the existing evaluation metrics in subject-driven generation. Now, with your valuable suggestion, we are delighted to present a more reliable and sound metric as mentioned in the response to your question ***W4-1: Evaluation on identity preservation***, and it shows the superiority of our method on identity preservation compared with current state-of-the-arts.
>
>
> ---
>
> We sincerely thank you once again for your valuable comments, which have greatly helped improve our work. We hope that the explanation above can well address your concerns!

---

> > ### Author Response · Authors · 2025-08-05
> >
> > Dear Reviewer HPe1,
> >
> > We would like to once again thank you for your valuable review and thoughtful questions!
> >
> > As the discussion period is coming close to an end, we would like to send you a reminder about our rebuttal as above to solve your concerns. Please check whether your concerns have been addressed. We are sincerely looking forward to hearing from you, and are always happy to have more discussions with you!
> >
> > Thank you once again for your insightful reviews!
> >
> > Best,
> >
> > Authors

---

> > ### Comment · Reviewer_HPe1 · 2025-08-06
> > **Additional Clarification on Some Points**
> >
> > I would like to thank the authors for their detailed response:
> >
> > 1. Thank you for clarifying what context you are using the terms "identity preservation" and "personalization" in. However, I don't think that this usage is correct or standard, especially as you refer to examples of non-animate objects such as houses and traffic cones in your rebuttal ("W2: More diverse samples") for which the term "identity" seems confusing. In my opinion view/geometric consistency seems the more appropriate description. Please share why you think "identity preservation" is a better choice?
> >
> > 2. It is difficult to judge the quality of the results from a text description. I'm not convinced by the argument that animals are the most common objects used in dynamic scenes. The proposed method claims to work for single image-to-3D tasks as well, so the results don't have to be limited to dynamic scenes. In fact, the method compares to Hunyuan3D and Trellis which show a much more diverse set of results.
> >
> > 3. Thank you for pointing out the 4D reconstruction results? However, I do not find any details about how a 4D reconstruction is obtained in the paper. Figure 2 provides an overview of a video/single-image to 3D process. Could you provide some additional details, or point me to the part of the paper that discusses this?
> >
> > 4. Thank you for the additional experiments! Could you provide some additional details about what exact metric you are measuring here, and what the experimental setting is (datasets, task, etc?).

---

> > > ### Author Response · Authors · 2025-08-09
> > > **Reply to Additional Clarification for Reviewer HPe1 [1/2]**
> > >
> > > Dear Reviewer HPe1,
> > >
> > > We sincerely thank you for the detailed comments! We would like to give more clarifications / justifications on your questions:
> > >
> > > ---
> > >
> > > 1. First, we would like to kindly point out that "identity preservation" and "personalization" are commonly used terminologies ***even for non-animate objects*** in subject-driven generation, which can be referred in the original DreamBooth [a] paper that also use these terminologies on non-animate subjects. Second, comparing with the task description of view/geometric consistency, the reason we define our task as "subject-driven 3D/4D generation for identity preservation" is that even if the generated assets achieve superior appearance/geometry consistency across different views, the assets may still not well capture the identity of the subject:
> > > > - For 3D assets with only good geometry consistency, the visualization in Fig. 1 is a representative example for the issue, where the side views and back views could suffer from inconsistent appearance and texture.
> > > > - For 3D assets with only decent appearance consistency, it could also be problematic (also explained in the most recent reply to Reviewer jNcb). Many methods including the most recent one Hunyuan3D-2.5 have discrepancies of the front view with the given image, such as different textures and over-smoothed appearance. The appearance consistency only enforces the consistency of all the generated views, but does not guarantee the generated views can match the characteristics of the given source image of the subject.
> > >
> > > Therefore, we think that "subject-driven generation with identity preservation" is the most proper task description towards our goal.
> > >
> > > ---
> > >
> > > 2. Yes, we agree with the reviewer that we should show more examples on more diverse scenes, and we will add these examples like what we described in our rebuttal in the final version. Our method can definitely work on more general objects and scenes, because there is no difference or specific challenges in handling general objects compared with handling scenes with animals. Moreover, we conduct a quantitative evaluation as below on a set of 136 view samples on the in-the-wild data we experiment on (the samples shown in L4GM [b] official website, as described in Lines 248-249) which contains more general objects and scenes, using the same evaluation way as the rebuttal to your question **W4-1: Evaluation on identity preservation** to show the effectiveness of our method on more general scenes (although these scenes also contain animals in most cases, it has much more general objects involved in the scenes, and we choose these scenes for evaluation because we already have their results in hand):
> > >
> > >
> > > |Methods|GPT-4o ($\uparrow$)|OpenAI o4-mini ($\uparrow$)|Gemma 3 27B ($\uparrow$)|Gemini 2.0 Flash ($\uparrow$)|Qwen2.5-VL-7B ($\uparrow$)|Mistral-Small-3.1-24B-Instruct ($\uparrow$)|Average ($\uparrow$)|
> > > |:---|:---:|:----:|:----:|:----:|:----:|:----:|:---:|
> > > |L4GM | **2.743** | 2.441 | 2.640 | 2.022 | 2.104 | 2.125| 2.346  |
> > > |TIRE (Ours) on L4GM | 2.601 | **2.478** | **2.684** | **2.110** | **2.191** | **2.163** |**2.371**  |
> > >
> > >
> > > Additional details of this evaluation process is explained in the following part to your fourth question.
> > >
> > > ---
> > >
> > > [a] Ruiz et al. DreamBooth: Fine Tuning Text-to-Image Diffusion Models for Subject-Driven Generation. CVPR 2023.
> > >
> > > [b] Ren et al. L4GM: Large 4D Gaussian Reconstruction Model. NeurIPS 2024.
> > >
> > > ---
> > >
> > > *(continuing in the following post)*

---

> > > ### Author Response · Authors · 2025-08-09
> > > **Reply to Additional Clarification for Reviewer HPe1 [2/2]**
> > >
> > > 3. Thank you for raising this point! We have some discussions in Lines 694-705 in the appendix, to discuss why we treat 4D generation as a sequence of 3D generation in our setting. Therefore, we could process only the first frame of 4D generation with our approach, then propagate the appearance of the first frame to the rest of the frames using the original 4D generation approach to achieve temporal consistency. The reason is that we want to handle 3D and 4D generation in a unified way, so it can be more broadly applied to various 4D generation approaches (more discussions about this point in the response to *Reviewer aG8d's W2-2 & Q3: Temporal consistency for video-to-4D*). We do agree that the details for 4D generation are not made clear enough in our original manuscript, and we will clarify the details of how we perform 4D generation in our final version.
> > >
> > > ---
> > >
> > > 4. We follow the same evaluation way as DreamBench++ [c], which utilizes VLMs to provide scores on how the identity of the generated samples match the source images. In our case, we render viewpoints from the generated 3D assets for evaluation. The metrics are the average scores across all the view samples. The dataset we used in our original rebuttal table is the DreamBooth-Dynamic dataset curated from the original DreamBooth dataset in our paper, and the table above in the reply to your second question is using the data displayed in the official L4GM paper which contains more general objects.
> > >
> > > We would like to sincerely thank you once again for the valuable comments, which greatly helped improve our work! If you find that all your concerns have been addressed, we would truly appreciate it if you could kindly consider raising the score!
> > >
> > >
> > > Best,
> > >
> > > Authors
> > >
> > > ---
> > >
> > > [c] Peng et al. DreamBench++: A Human-Aligned Benchmark for Personalized Image Generation. ICLR 2025.

---

### Official Review · Reviewer_aG8d · 2025-07-03

**Clarity:** 4
**Significance:** 2
**Originality:** 3
**Rating:** 4
**Confidence:** 3

**Summary:**

This paper introduces a pipeline for enhancing the appearance preservation (personalization) in image(video) to 3D(4D). First a 3D(4D) is initialized using ImageDream (and Wonder3D), then three steps are performed: Track - visibility masks are created for the 3D/4D representation with respect to the source view using a video tracker. Inpaint - the masks are progressively inpainted using a personalized inpainting model (similar to RealFill). Resplat - the multi-view images are resplatted using LG(4)M to obtain a 3D(4D) gaussian splatting representation. The pipeline can be seen as an extension for LG(4)M. Experimentally, they compare their method with 6 img-3D models and 3 vid-4D models and also conduct a user study.

**Questions:**

+ Could you clarify whether the observed appearance improvements are due to the proposed inpainting module or simply the use of Wonder3D for 3DGS color features? An ablation without inpainting would help isolate its effect and increase the score.

+ Can you provide specific examples of inconsistencies that arise without progressive inpainting?
+ Given that each frame at timestep t is processed independently in the pipeline, how do you address potential temporal inconsistencies across frames in video sequences?
+ Does the “30% of the denoising schedule” correspond to setting the inpainting strength parameter to 0.3? If so, could this be clarified explicitly in the appendix?

**Ethical Concerns:**

["NO or VERY MINOR ethics concerns only"]

**Final Justification:**

The rebuttal was able to address my main concerns and i'm happy to raise the score accordingly. Please include that " the inpainting part indeed plays an indispensable role for achieving good results with decent identity preservation" in the final revision.

**Limitations:**

yes

**Quality:**

3

**Strengths And Weaknesses:**

Strengths:
1. The method improves the shadowy bias of L(4)GM (Figs. 4-5), and corrects some geometry artifacts (Fig. 6), without heavy training, relying only on light inpainter fine-tuning and off-the-shelf models. It can be applied on top of any image-to-multiview (the initialization) and multiview-to-3D (the resplatting) models.
2. Since 3D/4D generation typically relies on small and scarce 3D/4D datasets (excluding slow SDS-like methods), this approach effectively combines geometry from methods trained on 3D/4D data with appearance from strong 2D generative models.
3. Progressive personalized inpainting, combined with visibility masks from a video tracker, is a solid and interesting idea that could be useful in other contexts.

**Major weaknesses:**
1. Appendix F states that the authors use Wonder3D for 3DGS color features and ImageDream for geometry features, due to ImageDream’s poor color quality. This makes it unclear whether appearance improvements stem from the inpainting module - the main contribution, or simply from using Wonder3D as the image-to-multiview backbone for color. An ablation without inpainting is needed to isolate its impact.
2. From my understanding, progressive inpainting should not introduce inconsistencies in an image-to-3D setting. Could the authors provide examples of such inconsistencies and clarify how the 30% denoising steps in the image-to-multiview model addresses them? Additionally, since each video frame at timestep t is processed independently, wouldn’t this lead to temporal inconsistencies across frames?

Minor Weaknesses:
1. Regarding reproducibility: most inpainting pipelines use a strength parameter, and I believe “30% of the denoising schedule” corresponds to setting this parameter to 0.3. It would be helpful to clarify this explicitly in the appendix.
2. Line 202: “A photo of sks” — It would be more appropriate to denote the rare token as $\hat{V}$ in the main text, and provide the exact token in the appendix. Additionally, in prior work, rare tokens are often prepended to a general class name. Have you considered extracting a general class (e.g., via BLIP or an LLM) to guide the inpainter more effectively?

Typos:
           Eq. 7, I believe that term should include a squared norm, please verify.

---

> ### Author Rebuttal · Authors · 2025-07-31
>
> We are sincerely grateful for your constructive feedback! We provide the following clarifications in response to your concerns:
>
> ---
>
> **W1 & Q1: Ablation without inpainting**
>
> Thanks for expressing the concern! Without inpainting, the 3D assets would largely be unready, which are caused by mainly two reasons:
>
> **(1)** The color component from Wonder3D mismatches with the underlying geometry of the 3D assets, which results in incorrect color fulfilling and messy textures for many parts of the 3D assets.
>
> **(2)** The color from Wonder3D is only for guidance and still away from good identity preservation quality. Even for the most recent 3D methods like Hunyuan3D-2.5, the identity preservation still needs to be improved.
>
>
> For this ablation, we have prepared very interesting qualitative results for the reviewers. However, due to the policy of the rebuttal not allowing for PDF or other types of media, we choose to describe the qualitative comparisons in detail to the reviewer.
>
>
> > ***Description of Visual Comparisons***
>
> > We use the "dog eating apple" example shown in Fig. 3 for easy reference. Without inpainting, there are many obvious flaws from the results. For example, on the right view of the scene, the dog's nose has a large grayish patch. The dog's body is also covered with messy dark brown patterns. Therefore, it is still far away from a good 3D asset for decent identity preservation. After the inpainting is done, the large grayish patch on the dog's nose gets infilled with reasonable color, and the textures on the dog's body gets more realistic and coherent with the given source view.
>
> Therefore, the inpainting part indeed plays an indispensable role for achieving good results with decent identity preservation. We will present this ablation result in the final version of our paper.
>
>
> ---
>
>
> **W2-1 & Q2: Inconsistencies introduced during inpainting and temporal consistency across frames**
>
> Thanks for the insightful comments! Your intuition is mostly correct! With our designed solution of progressive inpainting, we indeed manage to avoid the inconsistencies to a great extent. The remaining inconsistency is mainly caused by the inaccurate tracking results from the tracker. Once the tracker yields a false negative (a position in the target view should be tracked to a valid position in source view, but the tracker gets lost), the inconsistency emerges. Basically, we fail to propagate the pixel values from the source view to the positions of the false negative pixels in the target view. Consequently, since the pixel loses track from the source view, it will be wrongly marked as "inpainting needed", which results in inconsistent inpainted appearance.
>
> The false negatives are most commonly seen at the boundaries of the objects. As we can observe from Fig. 6 in the main paper and Fig. I in the appendix, the inconsistent geometry is most obvious around the object boundaries.
>
> ---
>
> **W2-2 & Q3: Temporal consistency for video-to-4D**
>
> In this work, we are mostly focusing on identity preservation of the generated 3D/4D assets, which is often a static attribute of the subjects regarding their appearance. Therefore, we do not have special designs tailored for temporal consistencies, but we observe that it can be decently handled with the base 4D generation models we adopt. The multiview-to-4D lifting model that we are using, L4GM, enforces temporal consistency from its training process. Specifically, for a given video sequence from the source view, together with the multi-view observations for the first frame generated by our method, the base 4D generation model is capable of generating multi-view dynamic videos with decent temporal consistency.
>
> Essentially, what we propose is a unified solution for both 3D and 4D generation, while for 4D generation we only need to generate multi-view observations for static assets, and pass the tasks of handling temporal consistency to the 4D base models. This unified and general solution grants our method more compatibility with the ever-evolving feed-forward 4D generation models (e.g., Splat4D [a], 4D-LRM [b]) in the future, and can benefit from their strong capability of maintaining temporal consistencies.
>
>
> [a] Yin et al. Splat4D: Diffusion-Enhanced 4D Gaussian Splatting for Temporally and Spatially Consistent Content Creation. SIGGRAPH 2025.
>
> [b] Ma et al. 4D-LRM: Large Space-Time Reconstruction Model From and To Any View at Any Time. arXiv:2506.18890.
>
> ---
>
> **W3 & Q4: Clarification on denoising schedule**
>
> Yes, the understanding from the reviewer is correct! 30% of the denoising schedule corresponds to setting the strength parameter in the inpainting pipeline to 0.3 in practice. We will clarify this explicitly in the final version of the paper. Thanks for the suggestion to make our paper more clear!
>
>
> ---
>
> **W4: Text prompts for inpainting**
>
> Thanks for the thoughtful comment! We believe that it is definitely a good idea to extract a general class name for guiding the inpainter. The reason for us to not have the general class name in our current version is that we follow the exact solution in RealFill [c], which also did not extract a general class name for guiding the inpainting process. We find this solution also generally works well in our setting, so we keep this simple and general solution in our work. Nevertheless, your comment is indeed very inspiring, as including the scene content or class name for the text prompt would definitely have the potential to further improve the inpainting performance! For the notation of the rare token, we will revise the corresponding text in our final version.
>
> [c] Tang et al. RealFill: Reference-Driven Generation for Authentic Image Completion. ACM TOG 2024.
>
> ---
>
> **T1: Typo**
>
> Thanks for the careful reading and spotting the typo! Yes, in Eq. 7, a squared norm should be included in the loss term!
>
>
> ---
>
>
> We would like to sincerely thank you once again for your valuable comments, which have greatly helped improve our work. We hope that the explanation above can well address your concerns!

---

> > ### Author Response · Authors · 2025-08-05
> >
> > Dear Reviewer aG8d,
> >
> > We would like to once again thank you for your valuable feedback and constructive questions!
> >
> > As the discussion period is coming close to an end, we would like to send you a reminder about our rebuttal as above to solve your concerns. Please check whether your concerns have been addressed. We are sincerely looking forward to hearing from you, and are always happy to have more discussions with you!
> >
> > Thank you once again for your insightful reviews!
> >
> > Best,
> >
> > Authors

---

### Official Review · Reviewer_jNcb · 2025-07-04

**Clarity:** 3
**Significance:** 1
**Originality:** 1
**Rating:** 4
**Confidence:** 4

**Summary:**

This paper tries to address the problem of 3D and 4D generation while preserving the subject identity. They first use off the shelf point tracker to find out regions that need to be inpainted and then rerun 3D reconstruction to obtain better 3D/4D generation. They compare the proposed pipeline with some existing 3D/4D reconstruction methods and show that it can indeed improve the unnatural back side views.

**Questions:**

- I am mainly concerned with the limited scope of this paper.. Authors motivates this method by showing that some methods generate unrealistic backside views which are not faithful to the input views. The example comes from LGM, but what is the real reason for this? As far as I know, this is due to the multi-view diffusion stage of ImageDream. A simple way to address this is to fine tune ImageDream with better images. If full model fine tuning is too expensieve, LoRA based fine tuning could also be considered. In this regard, the pipeline is designed to address a problem specific to one particular model. In order to actually show this method is general, can the authors demonstrate:
1. The backside artifact/identity not preserving problem is a general problem that lots of single image to 3D/4D generation models have? e.g. if the root cause of LGM is due to ImageDream, then one can simply address it by using another multi-view diffusion models. e.g. Zero123++. In fact Gaussian reconstruction model (GRM, Xu et al.) can take multi-views output from different models.
2. If the problem does exist in lots of models, can the authors show that the proposed pipeline is general to fix this type of problems? e.g. can this method also take other multi-view/3D generation models as input and improve their artefacts?



- Experiment evaluation is not solid. The paper claims to address both 3D and 4D generation. Both tasks are not properly evaluated and the results are not convincing.
1. For 3D, why is there only qualitative result? The selected examples from TRELLIS and HY3D do not look like the actual performance one would see for object reconstruction. Quantitative evaluation with at least 100 examples is needed, i.e. via CLIP score or user study, or more advanced 3D generation metrics (e.g. MET3R, CVPR'25).
2. For 4D, why is SV4D not compared is the user study, and also table 1? This is related to the first point, does SV3D/SV4D actually have the problem of artifacts in the backside views?

**Ethical Concerns:**

["NO or VERY MINOR ethics concerns only"]

**Final Justification:**

Before the rebuttal stage, I was mainly concerned that the approach from this paper focuses on a small problem and the pipeline is not general. Author response show that it is a common problem for lots of models and their pipeline does improve from them, by showing quantitative evaluation results. Therefore, I am willing to improve my score and urge the author to add the additional explaination and results to the revision.

**Limitations:**

yes

**Quality:**

2

**Strengths And Weaknesses:**

**Strength**
The paper is overall well written, it is easy to follow the whole pipeline.

**Weakness**
- Significance: the problem this paper tries to address seems to be very specific to LGM -- the backside views do not look natural and faithful to frontview, same for LG4M. The root cause of this is actually comes from ImageDream which generates blueish images in the backside views. I am not sure this is a really meaningful problem, as this might be largely due to data issue. hence can be trivially solved with better image renderings and fine tune the model.
- Originality. The whole pipeline is basically combining several blocks together and making the system work. There is not much technical contribution.
- Quality. The evaluation is not solid. it is not convincing that the image to 3D is actually better than TRELLIS or HY3D. only a few qualitative examples are shown.

---

> ### Author Rebuttal · Authors · 2025-07-31
>
> We sincerely appreciate the detailed feedback from you for our work! We provide the following clarifications in response to your concerns:
>
> ---
>
> **W1 & Q1-1: Significance and the scope of the paper**
>
> We would like to point out that the problem we are solving is ***not*** specific for certain models of LGM/L4GM. This issue still exists for other models, and even for the most recent 3D methods like Hunyuan3D-2.5. The blueish effect, as shown in Fig. 1, is just an example of how the generated views can look different from the identity of the source view. Solving the dataset bias, as suggested by the reviewer, is actually very challenging to accomplish. We have some arguments in Lines 651-655 about the infeasibility of enumerating every bias issue during dataset filtering.
>
> Take the most recent method Hunyuan3D-2.5 as an example, they already have careful dataset preprocessing steps for optimizing the appearance for back views and side views. However, if playing with the Hunyuan3D-2.5 demo with some in-the-wild images, we can observe the issue of the generated 3D assets having unnaturally smooth and textureless appearances, along with a whitish tone. It may be a result of the data filtering strategy used in Hunyuan3D-2.5, as the data cleaning algorithm filters out "bad" data according to certain rules. At the same time, it inevitably gets biased towards other issues that are the opposites of the data properties being filtered out.
>
> We have prepared intriguing qualitative results for the reviewers. However, due to the policy of the rebuttal not allowing for PDF or other types of media, we choose to describe the qualitative comparisons in detail to the reviewer. *The qualitative comparisons requested for the following questions are addressed in the same way.*
>
>
> > ***Description of Visual Comparisons***
>
> > For the most recent advancement of Hunyuan3D-2.5, we place 5 samples generated by the model side by side, and it becomes obvious to see that for the side views and back views of the object, although the shape and color tones looks generally reasonable, the appearance is over-smoothed and looks unrealistic. This could be the consequence of using certain rules for data filtering and preprocessing, and the rest of the data gets biased towards the opposite direction.
>
>
>
>
> Besides the dataset bias issues which are already very challenging to solve as mentioned above, there are other issues that result in identity mismatch of the generated back views and source views. For example, **(1)** the prior or bias introduced in the optimization target when training the 3D/4D generation model, and **(2)** the inaccurate boundaries of the reference view of the object, when segmenting the object out from an in-the-wild image/video. Therefore, resolving the issue of unfaithful generated novel view appearance is very challenging. Our paper provides a "use-as-needed" solution that can be applied whenever the user feels that the back and side views of the generated assets need to be enhanced. Thus, we believe that our paper can be broadly useful for various types of 3D generation methods to enhance their quality regarding identity preservation.
>
>
> ---
>
> **Q1-2: Can our model serve as a general solution to fix the problem?**
>
> Yes! This is actually a strength of our proposed method, as our solution is ***representation-agnostic***. It is because our progressive texture infilling is completely performing in 2D, leveraging the powerful 2D tools for the 3D task. Therefore, for handling any 3D generation methods with any 3D representations, our pipeline is the same, which demonstrates the general applicability and broad scope of our method. More specifically, we will first render views from the 3D representation, and then use *Track* to identify the infilling mask, *Inpaint* to infill the unseen regions, and *Resplat* to unproject the 2D observations back to 3D. Specifically, we choose Wonder3D which uses 3D mesh as the representation to demonstrate the general capability of our model. The description of the qualitative results is shown below:
>
>
> > ***Description of Visual Comparisons***
>
> > We use the "dog eating apple" example shown in Fig. 3 for easy reference. For the results from Wonder3D, the back and side views have weird dark brown patterns mixed with the orange color of the dog. We render from different viewpoints and apply our method to progressively inpaint the scene. Actually, 3D mesh as representation would even make the *Track* stage easier as 3D mesh generally has less floating artifacts than other representations like NeRFs or 3D Gaussians (more discussions on this point can be referred in **Reviewer AB6n's W1 & Q1: Which 3D representations are more suitable?**). After our method is applied, the weird dark brown pattern issue gets mitigated, and thus improves the identity preservation of the generated asset.
>
>
>
> ---
>
>
>
> **W2: Originality**
>
> The originality of our paper comes from our innovation to leverage the powerful 2D models like tracking and inpainting tools for 3D generation tasks. 2D models are trained on massive images and videos, which can come to rescue for the 3D field that has relatively scarce data. Also, as acknowledged by other reviewers, the utilization of strong 2D generative models (Reviewer aG8d) is novel and interesting (Reviewer HPe1). Specially, our creative backward tracking (Reviewer AB6n) and progressive personalized inpainting is a solid and interesting idea that can also be useful in other contexts (Reviewer aG8d).
>
> Also, seamlessly combining these methods is a non-trivial job. For example, the *Track* stage not only identifies the regions that need to be inpainted, but also propagates the corresponding pixels along the tracking path to novel views to mitigate the difficulty of the following *Inpaint* stage, and our backward tracking is designed for better inpainting quality in the *Inpaint* stage. Therefore, our three-stage pipeline is delicately designed and well orchestrated.
>
>
>
> ---
>
> **W3 & Q2-1: Quality comparison with existing methods**
>
> Thanks for the suggestions! Also suggested by Reviewer HPe1, we follow DreamBench++ [a] to adopt MLLM-based evaluation on identity preservation and the generation quality. Besides GPT-4o which is adopted in DreamBench++, we also use five other vision-language models (VLMs) with different architectures and sizes to foster the reliability of the evaluation. The results are reported below with 184 view samples:
>
>
> |Methods|GPT-4o ($\uparrow$)|OpenAI o4-mini ($\uparrow$)|Gemma 3 27B ($\uparrow$)|Gemini 2.0 Flash ($\uparrow$)|Qwen2.5-VL-7B ($\uparrow$)|Mistral-Small-3.1-24B-Instruct ($\uparrow$)|Average ($\uparrow$)|
> |:---|:---:|:----:|:----:|:----:|:----:|:----:|:---:|
> |TRELLIS|1.332|1.426|1.870|1.402|1.596|1.228|1.476|
> |Hunyuan3D-2.5 | 1.614 | 1.690 | 2.098 | 1.533 | 1.780 | 1.501| 1.703  |
> |TIRE (Ours) | **1.777** | **1.834** | **2.103** | **1.793** | **1.880** | **1.739** |**1.854**  |
>
> We can observe that **(1)** our method consistently outperforms the state-of-the-arts TRELLIS and Hunyuan3D-2.5 in the quantitative evaluation regarding identity preservation and generation quality; **(2)** different VLMs share consistent scoring rank for different models, demonstrating the robust generalization of the adopted evaluation protocol.
>
> [a] Peng et al. DreamBench++: A Human-Aligned Benchmark for Personalized Image Generation. ICLR 2025.
>
> ---
>
> **Q2-2: Comparison with SV4D**
>
> The reason for us to exclude SV4D in the user study and Tab. 1 in our original manuscript is that the official code of SV4D has only released the multi-view video generation part with 21 pre-defined viewpoints, without the following 4D optimization step. Therefore, only a certain number of selected views can be rendered with its official code, so we choose to pick the closest viewpoints only for qualitative comparison in the appendix. To resolve the concern, we also evaluate the DINO-I scores in the same setting as Tab. 1 based on these closest viewpoints.
>
> |Methods|DINO-I (ViT-S/16) ($\uparrow$)|DINO-I (ViT-B/16) ($\uparrow$)|
> |:---|:---:|:---:|
> |SV4D| 0.5213 |0.5426|
> |Ours| **0.5665** | **0.5815** |
>
> We can see that SV4D still has inferior performance than our method regarding identity preservation. For qualitative comparison with SV3D/SV4D, we have also prepared informative visual results for the reviewer, and the description of the visualizations is shown below:
>
>
> > ***Description of Visual Comparisons***
>
> > From a group of SV3D/SV4D results on side views and back views placed side by side, we can observe that they often suffer from whitish tone and blurry texture at the generated viewpoints, which may still originate from the imbalance object properties of the dataset used for training the model, or the bias from the network design. It indicates that the problem still exists for SV3D/SV4D.
>
> ---
>
> We would like to sincerely thank you once again for your valuable comments, which have greatly helped improve our work. We hope that the explanation above can well address your concerns!

---

> > ### Comment · Reviewer_jNcb · 2025-08-04
> > **thanks for the detailed response**
> >
> > I appreciate the authors for their detailed response.
> > However, it does not fully address my concerns.
> >
> > In W3 and Q2.1 and Q2.2, the authors provide further quantitative results which shows their method is better. But which model are you using now? this still does not address my concern: is the proposed pipeline a general approach to fix a diverse set of 2d/3D generation models? e.g. can I plug Hunyuan3d model into this pipeline and get even better results than hunyuan3d?

---

> > > ### Author Response · Authors · 2025-08-04
> > >
> > > Dear Reviewer jNcb,
> > >
> > > Thanks so much for your reply! For the results in W3, Q2.1, and Q2.2, the base models we were using are LGM for 3D generation, and L4GM for 4D generation, which are the same setting used throughout our paper. The comparisons in W3, Q2.1, and Q2.2 show that methods like SV4D, TRELLIS, Hunyuan3D-2.5 are still away from solving our task of subject-driven 3D/4D generation, demonstrating the significance of our research problem.
> > >
> > > For our proposed pipeline, it is a general approach to fix a diverse set of 3D generation models. Specifically, we only need to use the approach that we want to fix (e.g., Hunyuan3D) to generate a 3D asset in the setup stage, as mentioned in Lines 114-115 (corresponding to the setup stage at the leftmost part of Figure 2). Afterwards, we can apply our approach on the images rendered from the initial 3D asset. **In short, we can simply plug in the model we want to fix as the model used in the setup stage.** The reason for this benefit is that our progressive texture infilling is completely performing in 2D. It not only enables us to leverage the powerful 2D tools that are trained on massive image/video data, but also makes our method a general solution to all types of 3D generation methods, because we can always render 2D observations from any 3D models.
> > >
> > > Thank you once again for your constructive reviews and discussing with us! We sincerely hope that our clarification can fully address your concerns!
> > >
> > > Best regards,
> > >
> > > Authors

---

> > > > ### Comment · Reviewer_jNcb · 2025-08-06
> > > >
> > > > Thank you for the response.
> > > > I am still not fully convinced that this is really a general pipeline without showing that it actually improves over Hunyuan3D by using Hunyuan3d as the base setup model. e.g. though some quantitative evaluation similar to current tables in W3.
> > > > As the technical contribution of this paper is limited, it is important to thoroughly verify and evaluate the claim that this is a general pipeline, and it does work for different base models.

---

> > > > > ### Author Response · Authors · 2025-08-09
> > > > >
> > > > > Dear Reviewer jNcb,
> > > > >
> > > > > Thank you so much for the question! We have finally finished the experiments of applying our method on Hunyuan3D-2.5, and evaluated the performance in the same setting as the table presented in the reply to your question ***W3 & Q2-1: Quality comparison with existing methods***:
> > > > >
> > > > > |Methods|GPT-4o ($\uparrow$)|OpenAI o4-mini ($\uparrow$)|Gemma 3 27B ($\uparrow$)|Gemini 2.0 Flash ($\uparrow$)|Qwen2.5-VL-7B ($\uparrow$)|Mistral-Small-3.1-24B-Instruct ($\uparrow$)|Average ($\uparrow$)|
> > > > > |:---|:---:|:----:|:----:|:----:|:----:|:----:|:---:|
> > > > > |Hunyuan3D-2.5 | 1.614 | 1.690 | **2.098** | 1.533 | 1.780 | 1.501| 1.703  |
> > > > > |TIRE (Ours) on Hunyuan3D-2.5 | **1.750** | **1.712** | 2.092 | **1.609** | **1.899** | **1.707** |**1.795**  |
> > > > >
> > > > > We can observe that our method also enhances identity preservation of the most recent Hunyuan3D-2.5 model, which demonstrates that our method is a general solution applicable for various types of 3D generation methods. One thing you may find strange is that *TIRE (Ours) on Hunyuan3D-2.5* has inferior performance than *TIRE (Ours) on LGM* presented in the original rebuttal table. This is because the front views generated by Hunyuan3D-2.5 have some variations from the given source view, while the front views from LGM simply more precisely ***copy*** the characteristics of the source reference image. This is because Hunyuan3D-2.5 generally sacrifices a bit of the identity preservation of the front view in its initial results to get an overall better appearance across all the views, while the initial results of LGM has low cross-view consistency, though its front view matches more accurately to the source image. The accuracy of the front view can affect the identity preservation of the other views, because part of the appearance of the other views are propagated from the front view during the *Track* stage.
> > > > >
> > > > > Thank you once again for your valuable comments! We sincerely hope that our additional results can well address your concern on whether our method can be a general solution to all types of 3D generation methods!
> > > > >
> > > > > Best,
> > > > >
> > > > > Authors

---

> > > ### Author Response · Authors · 2025-08-05
> > >
> > > Dear Reviewer jNcb,
> > >
> > > Thank you again for your constructive reviews and discussion with us! As the discussion period is coming close to the end, we would like to hear from you about whether our follow-up response has addressed your remaining concerns.
> > >
> > > Please feel free to reach out if there is any remaining question, and we truly appreciate your great effort!
> > >
> > > Best regards,
> > >
> > > Authors

---

### Author Response · Authors · 2025-08-09
**Summary of Author-Reviewer Discussion**

Dear ACs and Reviewers,

We deeply appreciate your thoughtful feedback and constructive engagement, which greatly helped to improve our submission! As the reviewer-author discussion phase comes to an end, we would like to summarize the major insights and analyses from the results provided in the rebuttal and discussion:

- **Quantitative evaluation metrics.** We adopted the VLM-based scoring evaluation protocol following previous works on identity preservation, which demonstrates that our method indeed enhances the identity preservation of the generated 3D/4D assets. The evaluations are conducted on both our curated DreamBooth-Dynamic dataset and the in-the-wild data. *Details are in the rebuttal and reply for Reviewers jNcb and HPe1.*

- **Our method as a general solution for all types of 3D generation methods.** We explained both conceptually and experimentally on why our method can be generally applicable to various 3D generation methods. Conceptually, we simply need to use the selected method in the setup stage to generate the initial asset, and then render multi-view observations to start our three-stage method, which any 3D method can perform. Experimentally, we have described how different 3D representations have advantages and disadvantages in tracking and overall performance, and also showcase that our solution can provide benefit to the most recent method Hunyuan3D-2.5. *Details are in the rebuttal and reply for Reviewers jNcb and AB6n.*

- **Abundant ablations for providing guidance on the model usage.** We have provided numerous ablations including the necessity of the *Inpaint* stage, degrees of progressiveness during inpainting, different 3D representations, etc. to provide more comprehensive analysis and useful guidance on how to better leverage our approach. *Details are in the rebuttal and reply for Reviewers aG8d and AB6n.*

- **Additional demonstrations to support the motivation.** We have provided additional demonstrations on more diverse samples, the failure cases of existing methods, etc. to better justify the importance of our task and showcase the capability of our method. *Details can be found in the* ***Description of Visual Comparisons*** *parts in the rebuttals.*

Due to the policy of the rebuttal phase not allowing us to show visual content, we have described the visual content in detail, but we will include these intriguing visual comparisons and demonstrations in our final version.

We would like to express our appreciation once again for everyone's helpful comments and engagement in the discussion period!

Best,

Authors

---

### Note · Authors · 2025-08-12

Dear Reviewers, ACs, SACs,

We deeply appreciate your thoughtful feedback and constructive engagement, which helped to improve our submission! We would like to summarize the key contributions of our work here to highlight the significance of our paper.

**Key Contributions:**
- **Crucial issue spotted for current 3D/4D generation methods.** We identified a crucial issue that is largely overlooked in nowadays 3D/4D generation field: ***the identity mismatch of the generated 3D/4D assets compared to the reference image***. We have demonstrated that even the most advanced models today like Hunyuan3D-2.5 still suffer from this issue that cannot be easily fixed by data filtering. Raising the awareness of this identity mismatch issue can be beneficial for the community to get inspiration for how to design the next-generation 3D/4D generative models.
- **Unleashing the powerful 2D tools for the 3D/4D generation task.** We innovatively leverage the powerful 2D tools like tracking and inpainting models that are trained on massive image/video data on the 3D task, which comes to rescue for the data scarcity of the 3D field. Also, our coordination of these 2D tools is not simple concatenation. Instead, we delicately design mechanisms like backward tracking and mask-aware denoising for refining the geometry. These designs enable the prior stages to lower the difficulty of the later stages, and later stages to make up the potential errors occurred in the prior stages. Therefore, our three-stage *Track*, *Inpaint*, *Resplat* solution leveraging the 2D tools is well orchestrated.
- **Generally applicable to all types of 3D/4D generation methods.** Our solution is a plug-in solution that is generally applicable to all types of 3D/4D representations and methods. This benefit is a ***natural outcome*** from our method design of fully leveraging 2D tools to solve this 3D task. To apply on any 3D/4D methods, it is as simple as replacing the process of generating the initial assets in our pipeline with the 3D/4D generation that we want to improve upon. Experimental results in the original paper, together with the rebuttal have shown that our solution can provide ***universal benefit*** for all types of 3D/4D representations and methods.

Thank you so much once again for your helpful comments and engagement in the discussion period! The clarifications, analyses, and results during the rebuttal and discussion phases will be incorporated into our final version.

Best,

Authors

---

### Decision · Program_Chairs · 2025-09-17

**Decision:**

Accept (poster)

**Comment:**

This paper introduces a pipeline for 3D/4D generation with improved subject identity preservation and consistency across views. The approach leverages a tracker to detect invisible regions, a progressive inpainting module for filling in missing details, and a resplatting stage for final 3D/4D Gaussian reconstruction. The method is modular, lightweight, and can be applied on top of existing image-to-multiview and multiview-to-3D pipelines.

Strengths include the clear and lightweight pipeline and practical improvements in handling view-inconsistency. The problem setting is timely and relevant  to the broader community.

Weaknesses initially concerned the rigor of the experimental evaluation and quality of the results. However, those concerns were largely addressed in the authors' rebuttal, resulting in an overall positive consensus.

The technical contribution is not strong enough for a spotlight/oral but provided that the reviewers' feedback is carefully incorporated in the final version, I recommend accept as a poster.